# Constrained episodic reinforcement learning in concave-convex and knapsack settings

**Kianté Brantley**
University of Maryland
kdbrant@cs.umd.edu

**Miroslav Dudík**
Microsoft Research
mdudik@microsoft.com

**Thodoris Lykouris**
Microsoft Research
thlykour@microsoft.com

**Sobhan Miryoosefi**
Princeton University
miryoosefi@cs.princeton.edu

**Max Simchowitz**
UC Berkeley
msimchow@berkeley.edu

**Aleksandrs Slivkins**
Microsoft Research
slivkins@microsoft.com

**Wen Sun**
Cornell University
ws455@cornell.edu

## Abstract

We propose an algorithm for tabular episodic reinforcement learning (RL) with constraints. We provide a modular analysis with strong theoretical guarantees for two general settings. First is the convex-concave setting: maximization of a concave reward function subject to constraints that expected values of some vector quantities (such as the use of unsafe actions) lie in a convex set. Second is the knapsack setting: maximization of reward subject to the constraint that the total consumption of any of the specified resources does not exceed specified levels during the whole learning process. Previous work in constrained RL is limited to linear expectation constraints (a special case of convex-concave setting), or focuses on feasibility question, or on single-episode settings. Our experiments demonstrate that the proposed algorithm significantly outperforms these approaches in constrained episodic benchmarks.

## 1 Introduction

Standard reinforcement learning (RL) approaches seek to maximize a scalar reward (Sutton and Barto, 1998, 2018; Schulman et al., 2015; Mnih et al., 2015), but in many settings this is insufficient, because the desired properties of the agent behavior are better described using constraints. For example, an autonomous vehicle should not only get to the destination, but should also respect safety, fuel efficiency, and human comfort constraints along the way (Le et al., 2019); a robot should not only fulfill its task, but should also control its wear and tear, for example, by limiting the torque exerted on its motors (Tessler et al., 2019). Moreover, in many settings, we wish to satisfy such constraints already during *training* and not only during the *deployment*. For example, a power grid, an autonomous vehicle, or a real robotic hardware should avoid costly failures, where the hardware is damaged or humans are harmed, already during training (Leike et al., 2017; Ray et al., 2020). Constraints are also key in additional sequential decision making applications, such as dynamic pricing with limited supply (e.g., Besbes and Zeevi, 2009; Babaioff et al., 2015), scheduling of resources on a computer cluster (Mao et al., 2016), and imitation learning, where the goal is to stay close to an expert behavior (Syed and Schapire, 2007; Ziebart et al., 2008; Sun et al., 2019).

In this paper we study *constrained episodic reinforcement learning*, which encompasses all of these applications. An important characteristic of our approach, distinguishing it from previous work (e.g., Altman, 1999; Achiam et al., 2017; Tessler et al., 2019; Miryoosefi et al., 2019; Ray et al., 2020), is our focus on *efficient exploration*, leading to reduced sample complexity. Notably, the modularity of

our approach enables extensions to more complex settings such as (i) maximizing concave objectives under convex constraints, and (ii) reinforcement learning under hard constraints, where the learner has to stop when some constraint is violated (e.g., a car runs out of gas). For these extensions, which we refer to as *concave-convex setting* and *knapsack setting*, we provide the first regret guarantees in the episodic setting (see related work below for a detailed comparison). Moreover, our guarantees are *anytime*, meaning that the constraint violations are bounded at any point during learning, even if the learning process is interrupted. This is important for those applications where the system continues to learn after it is deployed.

Our approach relies on the principle of *optimism under uncertainty* to efficiently explore. Our learning algorithms optimize their actions with respect to a model based on the empirical statistics, while optimistically overestimating rewards and underestimating the resource consumption (i.e., overestimating the distance from the constraint). This idea was previously introduced in multi-armed bandits (Agrawal and Devanur, 2014); extending it to episodic reinforcement learning poses additional challenges since the policy space is exponential in the episode horizon. Circumventing these challenges, we provide a modular way to analyze this approach in the basic setting where both rewards and constraints are linear (Section 3) and then transfer this result to the more complicated concave-convex and knapsack settings (Sections 4 and 5). We empirically compare our approach with the only previous works that can handle convex constraints and show that our algorithmic innovations lead to significant empirical improvements (Section 6).

**Related work.** Sample-efficient exploration in constrained episodic reinforcement learning has only recently started to receive attention. Most previous works on episodic reinforcement learning focus on unconstrained settings (Jaksch et al., 2010; Azar et al., 2017; Dann et al., 2017). A notable exception is the work of Cheung (2019) and Tarbouriech and Lazaric (2019). Both of these works consider vectorial feedback and aggregate reward functions, and provide theoretical guarantees for the reinforcement learning setting with a single episode, but require a strong reachability or communication assumption, which is not needed in the episodic setting studied here. Also, compared to Cheung (2019), our results for the knapsack setting allow for a significantly smaller budget, as we illustrate in Section 5. Moreover, our approach is based on a tighter bonus, which leads to a superior empirical performance (see Section 6). Recently, there have also been several concurrent and independent works on sample-efficient exploration for reinforcement learning with constraints (Singh et al., 2020; Efroni et al., 2020; Qiu et al., 2020; Ding et al., 2020; Zheng and Ratliff, 2020). Unlike our work, all of these approaches focus on linear reward objective and linear constraints and do not handle the concave-convex and knapsack settings that we consider.

Constrained reinforcement learning has also been studied in settings that do not focus on sample-efficient exploration (Achiam et al., 2017; Tessler et al., 2019; Miryoosefi et al., 2019). Among these, only Miryoosefi et al. (2019) handle convex constraints, albeit without a reward objective (they solve the feasibility problem). Since these works do not focus on sample-efficient exploration, their performance drastically deteriorates when the task requires exploration (as we show in Section 6).

Sample-efficient exploration under constraints has been studied in multi-armed bandits, starting with a line of work on dynamic pricing with limited supply (Besbes and Zeevi, 2009, 2011; Babaioff et al., 2015; Wang et al., 2014). A general setting for bandits with global knapsack constraints (*bandits with knapsacks*) was defined and solved by Badanidiyuru et al. (2018) (see also Ch. 10 of Slivkins, 2019). Within this literature, the closest to ours is the work of Agrawal and Devanur (2014), who study bandits with concave objectives and convex constraints. Our work is directly inspired by theirs and lifts their techniques to the more general episodic reinforcement learning setting.

## 2   Model and preliminaries

In episodic reinforcement learning, a learner repeatedly interacts with an environment across $K$ episodes. The environment includes the state space $\mathcal{S}$, the action space $\mathcal{A}$, the episode horizon $H$, and the initial state $s_0$.[1] To capture constrained settings, the environment includes a set $\mathcal{D}$ of $d$ resources where each $i \in \mathcal{D}$ has a capacity constraint $\xi(i) \in \mathbb{R}^+$. The above are fixed and known to the learner.

**Constrained Markov decision process.** We work with MDPs that have resource consumption in addition to rewards. Formally, a *constrained* MDP (CMDP) is a triple $\mathcal{M} = (p, r, \boldsymbol{c})$ that describes transition probabilities $p : \mathcal{S} \times \mathcal{A} \to \Delta(\mathcal{S})$, rewards $r : \mathcal{S} \times \mathcal{A} \to [0, 1]$, and resource consumption $\boldsymbol{c} : \mathcal{S} \times \mathcal{A} \to [0, 1]^d$. For convenience, we denote $c(s, a, i) = c_i(s, a)$. We allow stochastic rewards and consumptions, in which case $r$ and $\boldsymbol{c}$ refer to the conditional expectations, conditioned on $s$ and $a$ (our definitions and algorithms are based on this conditional expectation rather than the full conditional distribution).

We use the above definition to describe two kinds of CMDPs. The *true* CMDP $\mathcal{M}^\star = (p^\star, r^\star, \boldsymbol{c}^\star)$ is fixed but *unknown* to the learner. Selecting action $a$ at state $s$ results in rewards and consumptions drawn from (possibly correlated) distributions with means $r^\star(s, a)$ and $\boldsymbol{c}^\star(s, a)$ and supports in $[0, 1]$ and $[0, 1]^d$ respectively. Next states are generated from transition probabilities $p^\star(s, a)$. The second kind of CMDP arises in our algorithm, which is model-based and at episode $k$ uses a CMDP $\mathcal{M}^{(k)}$.

**Episodic reinforcement learning protocol.** At episode $k \in [K]$, the learner commits to a policy $\pi_k = (\pi_{k,h})_{h=1}^{H}$ where $\pi_{k,h} : \mathcal{S} \to \Delta(\mathcal{A})$ specifies how to select actions at step $h$ for every state. The learner starts from state $s_{k,1} = s_0$. At step $h = 1, \ldots, H$, she selects an action $a_{k,h} \sim \pi_{k,h}(s_{k,h})$. The learner earns reward $r_{k,h}$ and suffers consumption $\boldsymbol{c}_{k,h}$, both drawn from the true CMDP $\mathcal{M}^\star$ on state-action pair $(s_{k,h}, a_{k,h})$ as described above, and transitions to state $s_{k,h+1} \sim p^\star(s_{k,h}, a_{k,h})$.

**Objectives.** In the basic setting (Section 3), the learner wishes to maximize reward while respecting the consumption constraints in expectation by competing favorably against the following benchmark:

$$\max_{\pi} \mathbb{E}^{\pi, p^\star} \Big[ \sum_{h=1}^{H} r^\star(s_h, a_h) \Big] \qquad \text{s.t.} \qquad \forall i \in \mathcal{D} : \mathbb{E}^{\pi, p^\star} \Big[ \sum_{h=1}^{H} c^\star(s_h, a_h, i) \Big] \leq \xi(i), \qquad (1)$$

where $\mathbb{E}^{\pi, p}$ denotes the expectation over the run of policy $\pi$ according to transitions $p$, and $s_h, a_h$ are the induced random state-action pairs. We denote by $\pi^\star$ the policy that maximizes this objective.

For the basic setting, we track two performance measures: *reward regret* compares the learner's total reward to the benchmark and *consumption regret* bounds excess in resource consumption:

$$\text{REWREG}(k) \coloneqq \mathbb{E}^{\pi^\star, p^\star} \Big[ \sum_{h=1}^{H} r^\star(s_h, a_h) \Big] - \frac{1}{k} \sum_{t=1}^{k} \mathbb{E}^{\pi_t, p^\star} \Big[ \sum_{h=1}^{H} r^\star(s_h, a_h) \Big], \qquad (2)$$

$$\text{CONSREG}(k) \coloneqq \max_{i \in \mathcal{D}} \Big( \frac{1}{k} \sum_{t=1}^{k} \mathbb{E}^{\pi_t, p^\star} \Big[ \sum_{h=1}^{H} c^\star(s_h, a_h, i) \Big] - \xi(i) \Big). \qquad (3)$$

Our guarantees are *anytime*, i.e., they hold at any episode $k$ and not only after the last episode.

We also consider two extensions. In Section 4, we consider a concave reward objective and convex consumption constraints. In Section 5, we require consumption constraints to be satisfied with high probability under a cumulative budget across all $K$ episodes, rather than in expectation in a single episode.

**Tabular MDPs.** We assume that the state space $\mathcal{S}$ and the action space $\mathcal{A}$ are finite (tabular setting). We construct standard empirical estimates separately for each state-action pair $(s, a)$, using the learner's observations up to and not including a given episode $k$. Eqs. (4–7) define sample counts, empirical transition probabilities, empirical rewards, and empirical resource consumption.[2]

$$N_k(s, a) = \max \Big\{ 1, \sum_{t \in [k-1], \, h \in [H]} \mathbf{1}\{s_{t,h} = s, a_{t,h} = a\} \Big\}, \qquad (4)$$

$$\widehat{p}_k(s'|s, a) = \frac{1}{N_k(s, a)} \sum_{t \in [k-1], \, h \in [H]} \mathbf{1}\{s_{t,h} = s, a_{t,h} = a, s_{t,h+1} = s'\}, \qquad (5)$$

$$\widehat{r}_k(s, a) = \frac{1}{N_k(s, a)} \sum_{t \in [k-1], \, h \in [H]} r_{t,h} \cdot \mathbf{1}\{s_{t,h} = s, a_{t,h} = a\}, \qquad (6)$$

$$\widehat{c}_k(s, a, i) = \frac{1}{N_k(s, a)} \sum_{t \in [k-1], \, h \in [H]} c_{t,h,i} \cdot \mathbf{1}\{s_{t,h} = s, a_{t,h} = a\} \quad \forall i \in \mathcal{D}. \qquad (7)$$

**Preliminaries for theoretical analysis.** The *Q-function* is a standard object in RL that tracks the learner's expected performance if she starts from state $s \in \mathcal{S}$ at step $h$, selects action $a \in \mathcal{A}$, and then follows a policy $\pi$ under a model with transitions $p$ for the remainder of the episode. We parameterize it by the *objective function* $m : \mathcal{S} \times \mathcal{A} \to [0,1]$, which can be either a reward, i.e., $m(s,a) = r(s,a)$, or consumption of some resource $i \in \mathcal{D}$, i.e., $m(s,a) = c(s,a,i)$. (For the unconstrained setting, the objective is the reward.) The performance of the policy in a particular step $h$ is evaluated by the value function $V$ which corresponds to the expected $Q$-function of the selected action (where the expectation is taken over the possibly randomized action selection of $\pi$). The $Q$ and value functions can be both recursively defined by dynamic programming:

$$Q_m^{\pi,p}(s,a,h) = m(s,a) + \sum_{s' \in \mathcal{S}} p(s'|s,a) V_m^{\pi,p}(s',h+1),$$

$$V_m^{\pi,p}(s,h) = \mathbb{E}_{a \sim \pi(\cdot|s)}\Big[ Q_m^{\pi,p}(s,a,h) \Big] \quad \text{and} \quad V_m^{\pi,p}(s,H+1) = 0.$$

By slight abuse of notation, for $m \in \{r\} \cup \{c_i\}_{i \in \mathcal{D}}$, we denote by $m^\star \in \{r^\star\} \cup \{c_i^\star\}_{i \in \mathcal{D}}$ the corresponding objectives with respect to the rewards and consumptions of the true CMDP $\mathcal{M}^\star$. For objectives $m^\star$ and transitions $p^\star$, the above are the *Bellman equations* of the system (Bellman, 1957).

Estimating the $Q$-function based on the model parameters $p$ and $m$ rather than the ground truth parameters $p^\star$ and $m^\star$ introduces errors. These errors are localized across stages by the notion of *Bellman error* which contrasts the performance of policy $\pi$ starting from stage $h$ under the model parameters to a benchmark that behaves according to the model parameters starting from the next stage $h+1$ but uses the true parameters of the system in stage $h$. More formally, for objective $m$:

$$\text{BELL}_m^{\pi,p}(s,a,h) = Q_m^{\pi,p}(s,a,h) - \Big( m^\star(s,a) + \sum_{s' \in \mathcal{S}} p^\star(s'|s,a) V_m^{\pi,p}(s',h+1) \Big). \tag{8}$$

Note that when the CMDP is $\mathcal{M}^\star$ ($m = m^\star$, $p = p^\star$), there is no mismatch and $\text{BELL}_{m^\star}^{\pi,p^\star} = 0$.

## 3 Warm-up algorithm and analysis in the basic setting

In this section, we introduce a simple algorithm that allows to simultaneously bound reward and consumption regrets for the basic setting introduced in the previous section. Even in this basic setting, we provide the first sample-efficient guarantees in constrained episodic reinforcement learning.[3] The modular analysis of the guarantees also allows us to subsequently extend (in Sections 4 and 5) the algorithm and guarantees to the more general concave-convex and knapsack settings.

**Our algorithm.** At episode $k$, we construct an estimated CMDP $\mathcal{M}^{(k)} = \big(p^{(k)}, r^{(k)}, \boldsymbol{c}^{(k)}\big)$ based on the observations collected so far. The estimates are *bonus-enhanced* (formalized below) to encourage more targeted exploration. Our algorithm CONRL selects a policy $\pi_k$ by solving the following constrained optimization problem which we refer to as BASICCONPLANNER($p^{(k)}, r^{(k)}, \boldsymbol{c}^{(k)}$):

$$\max_\pi \mathbb{E}^{\pi,p^{(k)}} \Big[ \sum_{h=1}^{H} r^{(k)}\big(s_h, a_h\big) \Big] \qquad \text{s.t.} \qquad \forall i \in \mathcal{D} : \mathbb{E}^{\pi,p^{(k)}} \Big[ \sum_{h=1}^{H} c^{(k)}\big(s_h, a_h, i\big) \Big] \le \xi(i).$$

The above optimization problem is similar to the objective (1) but uses the estimated model instead of the (unknown to the learner) true model. We also note that this optimization problem can be optimally solved as it is a linear program on the occupation measures (Puterman, 2014), i.e., setting as variables the probability of each state-action pair and imposing flow conservation constraints with respect to the transitions. This program is described in Appendix A.1.

**Bonus-enhanced model.** A standard approach to implement the principle of optimism under uncertainty is to introduce, at each episode $k$, a *bonus term* $\widehat{b}_k(s,a)$ that favors under-explored actions. Specifically, we add this bonus to the empirical rewards (6), and subtract it from the consumptions (7): $r^{(k)}(s,a) = \widehat{r}_k(s,a) + \widehat{b}_k(s,a)$ and $c^{(k)}(s,a,i) = \widehat{c}_k(s,a,i) - \widehat{b}_k(s,a)$ for each resource $i$.

Following the unconstrained analogues (Azar et al., 2017; Dann et al., 2017), we define the bonus as:

$$\widehat{b}_k(s,a) = H\sqrt{\frac{2\ln\left(8SAH(d+1)k^2/\delta\right)}{N_k(s,a)}}, \tag{9}$$

where $\delta > 0$ is the desired failure probability of the algorithm and $N_k(s,a)$ is the number of times $(s,a)$ pair is visited, c.f. (4), $S = |\mathcal{S}|$, and $A = |\mathcal{A}|$. Thus, under-explored actions have a larger bonus, and therefore appear more appealing to the planner. For estimated transition probabilities, we just use the empirical averages (5): $p^{(k)}(s'|s,a) = \widehat{p}(s'|s,a)$.

**Valid bonus and Bellman-error decomposition.** For a bonus-enhanced model to achieve effective exploration, the resulting bonuses need to be *valid*, i.e., they should ensure that the estimated rewards overestimate the true rewards and the estimated consumptions underestimate the true consumptions.

**Definition 3.1.** A bonus $b_k : \mathcal{S} \times \mathcal{A} \to \mathbb{R}$ is valid if, $\forall s \in \mathcal{S}, a \in \mathcal{A}, h \in [H], m \in \{r\} \cup \{c_i\}_{i \in \mathcal{D}}$:

$$\left|\left(\widehat{m}_k(s,a) - m^\star(s,a)\right) + \sum_{s' \in \mathcal{S}}\left(\widehat{p}_k(s'|s,a) - p^\star(s'|s,a)\right)V_{m^\star}^{\pi^\star,p^\star}(s',h+1)\right| \leq b_k(s,a).$$

By classical concentration bounds (Appendix B.1), the bonus $\widehat{b}_k$ of Eq. (9) satisfies this condition:

**Lemma 3.2.** *With probability $1 - \delta$, the bonus $\widehat{b}_k(s,a)$ is valid for all episodes $k$ simultaneously.*

Our algorithm solves the BASICCONPLANNER optimization problem based on a bonus-enhanced model. When the bonuses are valid, we can upper bound the per-episode regret by the expected sum of Bellman errors across steps. This is the first part in classical unconstrained analyses and the following proposition extends this decomposition to constrained episodic reinforcement learning. The proof uses the so-called simulation lemma (Kearns and Singh, 2002) and is provided in Appendix B.3.

**Proposition 3.3.** *If $\widehat{b}_k(s,a)$ is valid for all episodes $k$ simultaneously then the per-episode reward and consumption regrets can be upper bounded by the expected sum of Bellman errors (8):*

$$\mathbb{E}^{\pi^\star,p^\star}\left[\sum_{h=1}^{H} r^\star(s_h,a_h)\right] - \mathbb{E}^{\pi_k,p^\star}\left[\sum_{h=1}^{H} r^\star(s_h,a_h)\right] \leq \mathbb{E}^{\pi_k}\left[\sum_{h=1}^{H}\left|\text{BELL}_{r^{(k)}}^{\pi_k,p^{(k)}}(s_h,a_h,h)\right|\right] \tag{10}$$

$$\forall i \in \mathcal{D}: \qquad \mathbb{E}^{\pi_k,p^\star}\left[\sum_{h=1}^{H} c^\star(s_h,a_h,i)\right] - \xi(i) \leq \mathbb{E}^{\pi_k}\left[\sum_{h=1}^{H}\left|\text{BELL}_{c_i^{(k)}}^{\pi_k,p^{(k)}}(s_h,a_h,h)\right|\right]. \tag{11}$$

**Final guarantee.** One difficulty with directly bounding the Bellman error is that the value function is not independent of the draws forming $r^{(k)}(s,a)$, $\mathbf{c}^{(k)}(s,a)$, and $p^{(k)}(s'|s,a)$. Hence we cannot apply Hoeffding inequality directly. While Azar et al. (2017) propose a trick to get an $\mathcal{O}(\sqrt{S})$ bound on Bellman error in unconstrained settings, the trick relies on the crucial property of Bellman optimality: for an unconstrained MDP, its optimal policy $\pi^\star$ satisfies the condition, $V_{r^\star}^{\pi^\star}(s,h) \geq V_{r^\star}^{\pi}(s,h)$ for all $s, h, \pi$ (i.e., $\pi^\star$ is optimal at any state). However, when constraints exist, the optimal policy does not satisfy the Bellman optimality property. Indeed, we can only guarantee optimality with respect to the initial state distribution, i.e., $V_{r^\star}^{\pi^\star}(s_0,1) \geq V_{r^\star}^{\pi}(s_0,1)$ for any $\pi$, but not everywhere else. This illustrates a fundamental difference between constrained MDPs and unconstrained MDPs. Thus we cannot directly apply the trick from Azar et al. (2017). Instead we follow an alternative approach of bounding the value function via an $\epsilon$-net over the possible values. This analysis leads to a guarantee that is weaker by a factor of $\sqrt{S}$ than the unconstrained results. The proof is provided in Appendix B.6.

**Theorem 3.4.** *There exists an absolute constant $c \in \mathbb{R}^+$ such that, with probability at least $1 - 3\delta$, reward and consumption regrets are both upper bounded by:*

$$\frac{c}{\sqrt{k}} \cdot S\sqrt{AH^3} \cdot \sqrt{\ln(k)\ln\left(SAH(d+1)k/\delta\right)} + \frac{c}{k} \cdot S^{3/2}AH^2\sqrt{\ln\left(2SAH(d+1)k/\delta\right)}.$$

**Comparison to single-episode results.** In single-episode setting, Cheung (2019) achieves $\sqrt{S}$ dependency under the further assumption that the transitions are sparse, i.e., $\|p^\star(s,a)\|_0 \ll S$ for all $(s,a)$. We do not make such assumptions on the sparsity of the MDP and we note that the regret bound of Cheung (2019) scales linearly in $S$ when $\|p^\star(s,a)\|_0 = \Theta(S)$. Also, the single-episode setting requires a strong reachability assumption, not present in the episodic setting.

**Remark 3.5.** *The aforementioned regret bound can be turned into a PAC bound of $\tilde{\mathcal{O}}\left(\frac{S^2 A H^3}{\epsilon^2}\right)$ by taking the uniform mixture of policies $\pi_1, \pi_2, \ldots, \pi_k$.*

## 4  Concave-convex setting

We now extend the algorithm and guarantees derived for the basic setting to when the objective is concave function of the accumulated reward and the constraints are expressed as a convex function of the cumulative consumptions. Our approach is modular, seamlessly building on the basic setting.

**Setting and objective.** Formally, there is a concave reward-objective function $f : \mathbb{R} \to \mathbb{R}$ and a convex consumption-objective function $g : \mathbb{R}^d \to \mathbb{R}$; the only assumption is that these functions are $L$-Lipschitz for some constant $L$, i.e., $|f(x) - f(y)| \leq L|x - y|$ for any $x, y \in \mathbb{R}$, and $|g(x) - g(y)| \leq L\|x - y\|_1$ for any $x, y \in \mathbb{R}^d$. Analogous to (1), the learner wishes to compete against the following benchmark which can be viewed as a reinforcement learning variant of the benchmark used by Agrawal and Devanur (2014) in multi-armed bandits:

$$\max_{\pi} f\left(\mathbb{E}^{\pi, p^\star}\left[\sum_{h=1}^{H} r^\star(s_h, a_h)\right]\right) \quad \text{s.t.} \quad g\left(\mathbb{E}^{\pi, p^\star}\left[\sum_{h=1}^{H} \boldsymbol{c}^\star(s_h, a_h)\right]\right) \leq 0. \tag{12}$$

The reward and consumption regrets are therefore adapted to:

$$\textsc{ConvexRewReg}(k) := f\left(\mathbb{E}^{\pi^\star, p^\star}\left[\sum_{h=1}^{H} r^\star(s_h, a_h)\right]\right) - f\left(\frac{1}{k}\sum_{t=1}^{k} \mathbb{E}^{\pi_t, p^\star}\left[\sum_{h=1}^{H} r^\star(s_h, a_h)\right]\right),$$

$$\textsc{ConvexConsReg}(k) := g\left(\frac{1}{k}\sum_{t=1}^{k} \mathbb{E}^{\pi_t, p^\star}\left[\sum_{h=1}^{H} \boldsymbol{c}^\star(s_h, a_h)\right]\right).$$

**Our algorithm.** As in the basic setting, we wish to create a bonus-enhanced model and optimize over it. To model the transition probabilites, we use empirical estimates $p^{(k)} = \widehat{p}_k$ of Eq. (5) as before. However, since reward and consumption objectives are no longer monotone in the accumulated rewards and consumption respectively, it does not make sense to simply add or subtract $\widehat{b}_k$ (defined in Eq. 9) as we did before. Instead we compute the policy $\pi_k$ of episode $k$ together with the model by solving the following optimization problem which we call $\textsc{ConvexConPlanner}$:

$$\max_{\pi} \max_{r^{(k)} \in \left[\widehat{r}_k \pm \widehat{b}_k\right]} f\left(\mathbb{E}^{\pi, p^{(k)}}\left[\sum_{h=1}^{H} r^{(k)}(s_h, a_h)\right]\right) \text{ s.t.} \min_{\boldsymbol{c}^{(k)} \in \left[\widehat{c}_k \pm \widehat{b}_k \cdot \mathbf{1}\right]} g\left(\mathbb{E}^{\pi, p^{(k)}}\left[\sum_{h=1}^{H} \boldsymbol{c}^{(k)}(s_h, a_h)\right]\right) \leq 0.$$

The above problem is convex in the occupation measures,[4] i.e., the probability $\rho(s, a, h)$ that the learner is at state-action-step $(s, a, h)$ — c.f. Appendix A.2 for further discussion.

$$\max_{\rho} \max_{r \in \left[\widehat{r}_k \pm \widehat{b}_k\right]} f\left(\sum_{s,a,h} \rho(s, a, h) r(s, a)\right) \quad \text{s.t.} \min_{\boldsymbol{c} \in \left[\widehat{c}_k \pm \widehat{b}_k \cdot \mathbf{1}\right]} g\left(\sum_{s,a,h} \rho(s, a, h) \boldsymbol{c}(s, a)\right) \leq 0$$

$$\forall s', h : \quad \sum_{a} \rho(s', a, h+1) = \sum_{s,a} \rho(s, a, h) \widehat{p}_k(s'|s, a)$$

$$\forall s, a, h : \quad 0 \leq \rho(s, a, h) \leq 1 \quad \text{and} \quad \sum_{s,a} \rho(s, a, h) = 1.$$

**Guarantee for concave-convex setting.** To extend the guarantee of the basic setting to the concave-convex setting, we face an additional challenge: it is not immediately clear that the optimal policy $\pi^\star$ is feasible for the $\textsc{ConvexConPlanner}$ program because $\textsc{ConvexConPlanner}$ is defined with respect to the empirical transition probabilities $p^{(k)}$.[5] Moreover, when $H > 1$, it is not straightforward to show that objective in the used model is always greater than the one in the true model as the used

model transitions $p^{(k)}(s, a)$ can lead to different states than the ones encountered in the true model.[6] We deal with both of these issues by introducing a novel application of the mean-value theorem to show that $\pi^\star$ is indeed a feasible solution of that program and create a similar regret decomposition to Proposition 3.3 (see Proposition C.1 and more discussion in Appendix C.1); this allows us to plug in the results developed for the basic setting. The full proof is provided in Appendix C.

**Theorem 4.1.** *Let $L$ be the Lipschitz constant for $f$ and $g$ and let* REWREG *and* CONSREG *be the reward and consumption regrets for the basic setting (Theorem 3.4) with the failure probability $\delta$. With probability $1 - \delta$, our algorithm in the concave-convex setting has reward and consumption regret upper bounded by $L \cdot$ REWREG *and* $Ld \cdot$ CONSREG *respectively.*

The linear dependence on $d$ in the consumption regret above comes from the fact that we assume $g$ is Lipschitz under $\ell_1$ norm.

## 5   Knapsack setting

Our last technical section extends the algorithm and guarantee of the basic setting to scenarios where the constraints are hard which is in accordance with most of the literature on *bandits with knapsacks*. The goal here is to achieve aggregate reward regret that is sublinear in the time horizon (in our case, the number of episodes $K$), while also respecting budget constraints for as small budgets as possible. We derive guarantees in terms of *reward regret*, as defined previously, and then argue that our guarantee extends to the seemingly stronger benchmark of the best dynamic policy.

**Setting and objective.** Each resource $i \in \mathcal{D}$ has an aggregate budget $B_i$ that the learner should not exceed over $K$ episodes. Unlike the basic setting, where we track the consumption regret, here we view this as a hard constraint. As in most works on bandits with knapsacks, the algorithm is allowed to use a "null action" for an episode, i.e., an action that yields a zero reward and consumption when selected at the beginning of an episode. The learner wishes to maximize her aggregate reward while respecting these hard constraints. We reduce this problem to a specific variant of the basic problem (1) with $\xi(i) = \frac{B_i}{K}$. We modify the solution to (1) to take the null action if any constraint is violated and call the resulting benchmark $\pi^\star$. Note that $\pi^\star$ satisfies constraints in expectation. At the end of this section, we explain how our algorithm also competes against a benchmark that is required to respect constraints deterministically (i.e., with probability one across all episodes).

**Our algorithm.** In the basic setting of Section 3, we showed a reward regret guarantee and a consumption regret guarantee, proving that the average constraint violation is $\mathcal{O}(1/\sqrt{K})$. Now we seek a stronger guarantee: the learned policy needs to satisfy budget constraints with high probability. Our algorithm optimizes a mathematical program KNAPSACKCONPLANNER (13) that strengthens the consumption constraints:

$$\max_{\pi} \mathbb{E}^{\pi, p^{(k)}} \left[ \sum_{h=1}^{H} r^{(k)}(s_h, a_h) \right] \quad \text{s.t.} \quad \forall i \in \mathcal{D} : \mathbb{E}^{\pi, p^{(k)}} \left[ \sum_{h=1}^{H} c^{(k)}(s_h, a_h, i) \right] \leq \frac{(1 - \epsilon)B_i}{K}. \quad (13)$$

In the above, $p^{(k)}$, $r^{(k)}$, $\boldsymbol{c}^{(k)}$ are exactly as in the basic setting and $\epsilon > 0$ is instantiated in the theorem below. Note that the program (13) is feasible thanks to the existence of the null action. The following mixture policy induces a feasible solution: with probability $1 - \epsilon$, we play the optimal policy $\pi^\star$ for the entire episode; with probability $\epsilon$, we play the null action for the entire episode. Note that the above program can again be cast as a linear program in the occupancy measure space — c.f. Appendix A.3 for further discussion.

**Guarantee for knapsack setting.** The guarantee of the basic setting on this tighter mathematical program seamlessly transfers to a reward guarantee that does not violate the hard constraints.

**Theorem 5.1.** *Assume that $\min_i B_i \leq KH$, i.e., constraints are non-vacuous. Let* AGGREG$(\delta)$ *be a bound on the aggregate (across episodes) reward or consumption regret for the soft-constraint setting (Theorem 3.4) with the failure probability $\delta$. Let $\epsilon = \frac{\text{AGGREG}(\delta)}{\min_i B_i}$. If $\min_i B_i >$ AGGREG$(\delta)$ then, with probability $1 - \delta$, the reward regret in the hard-constraint setting is at most $\frac{2H \text{AGGREG}(\delta)}{\min_i B_i}$ and constraints are not violated.*

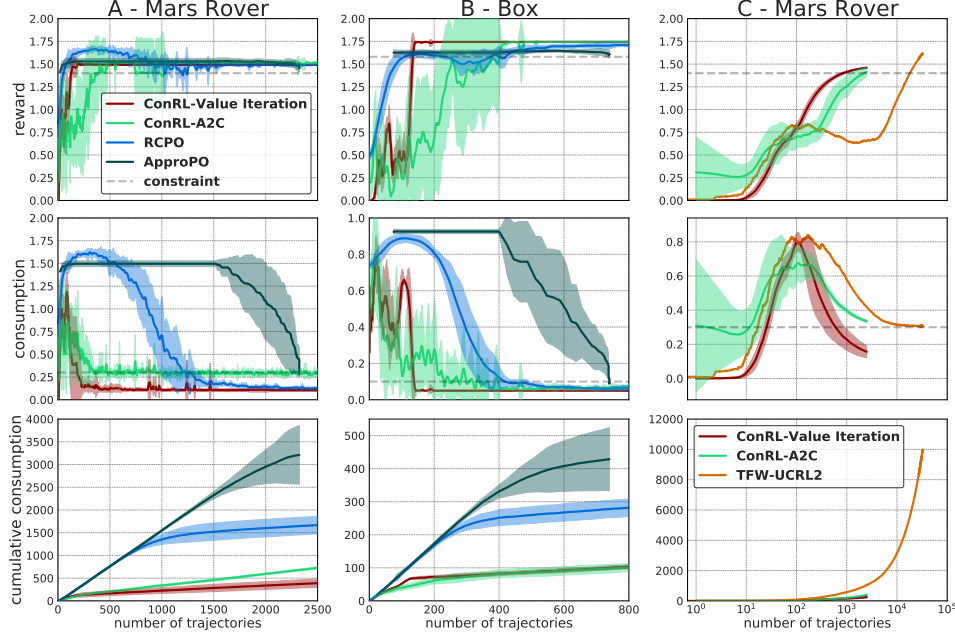

Figure 1: The performance of the algorithms as a function of the number of sample trajectories (trajectory = 30 samples); showing average and standard deviation over 10 runs. Dashed line in the second row is the upper bound on the consumption (for all algorithms), the dashed line in the first row is a lower bound on the reward (only required by APPROPO).

The above theorem implies that the aggregate reward regret is sublinear in $K$ as long as $\min_i B_i \gg H \text{AGGREG}(\delta)$. The analysis in the above main theorem (provided in Appendix D) is *modular* in the sense that it leverages the CONRL's performance to solve (13) in a black-box manner. Smaller $\text{AGGREG}(\delta)$ from the basic soft-constraint setting immediately translates to smaller reward regret and smaller budget regime (i.e., $\min_i B_i$ can be smaller). In particular, using the $\text{AGGREG}(\delta)$ bound of Theorem 3.4, the reward regret is sublinear as long as $\min_i B_i = \Omega(\sqrt{K})$.

In contrast, previous work of Cheung (2019) can only deal with larger budget regime, i.e., $\min_i B_i = \Omega(K^{2/3})$. Although the guarantees are not directly comparable as the latter is for the single-episode setting, which requires further reachability assumptions, the budget we can handle is significantly smaller and in the next section we show that our algorithm has superior empirical performance in episodic settings even when such assumptions are granted.

**Dynamic policy benchmark.** The common benchmark used in bandits with knapsacks is not the best stationary policy $\pi^\star$ that respects constraints in expectation but rather the best *dynamic* policy (i.e., a policy that makes decisions based on the history) that never violates hard constraints *deterministically*. In Appendix D, we show that the optimal dynamic policy (formally defined there) has reward less than policy $\pi^\star$ (informally, this is because $\pi^\star$ respects constraints in expectation while the dynamic policy has to satisfy constraints deterministically) and therefore the guarantee of Theorem 5.1 also applies against the optimal dynamic policy.

## 6 Empirical comparison to other concave-convex approaches

In this section, we evaluate the performance of CONRL against previous approaches.[7] Although our CONPLANNER (see Appendix A) can be solved exactly using linear programming (Altman, 1999), in our experiments, it suffices to use Lagrangian heuristic, denoted as LAGRCONPLANNER (see Appendix E.1). This Lagrangian heuristic only needs a planner for the *unconstrained* RL task. We consider two unconstrained RL algorithms as planners: value iteration and a model-based Advantage Actor-Critic (A2C) (Mnih et al., 2016) (based on fictitious samples drawn from the model provided as an input). The resulting variants of LAGRCONPLANNER are denoted CONRL-VALUE ITERATION

and CONRL-A2C. We run our experiments on two grid-world environments *Mars rover* (Tessler et al., 2019) and *Box* (Leike et al., 2017).[8]

**Mars rover.** The agent must move from the initial position to the goal without crashing into rocks. If the agent reaches the goal or crashes into a rock it will stay in that cell for the remainder of the episode. Reward is 1 when the agent reaches the goal and $1/H$ afterwards. Consumption is 1 when the agent crashes into a rock and $1/H$ afterwards. The episode horizon $H$ is 30 and the agent's action is perturbed with probability $0.1$ to a random action.

**Box.** The agent must move a box from the initial position to the goal while avoiding corners (cells adjacent to at least two walls). If the agent reaches the goal it stays in that cell for the remainder of the episode. Reward is 1 when agent reaches the goal for the first time and $1/H$ afterwards; consumption is $1/H$ whenever the box is in a corner. Horizon $H$ is 30 and the agent's action is perturbed with probability $0.1$ to a random action.

We compare CONRL to previous constrained approaches (derived for either episodic or single-episode settings) in Figure 1. We keep track of three metrics: episode-level reward and consumption (the first two rows) and cumulative consumption (the third row). Episode-level metrics are based on the most recent episode in the first two columns, i.e., we plot $\mathbb{E}^{\pi_k}[\sum_{h=1}^H r_h^\star]$ and $\mathbb{E}^{\pi_k}[\sum_{h=1}^H c_h^\star]$. In the third column, we plot the average across episodes so far, i.e., $\frac{1}{k}\sum_{t=1}^k \mathbb{E}^{\pi_t}[\sum_{h=1}^H r_h^\star]$ and $\frac{1}{k}\sum_{t=1}^k \mathbb{E}^{\pi_t}[\sum_{h=1}^H c_h^\star]$, and we use the log scale for the $x$-axis. The cumulative consumption is $\sum_{t=1}^k \sum_{h=1}^H c_{t,h}$ in all columns. See Appendix E for further details about experiments.

**Episodic setting.** We first compare our algorithms to two episodic RL approaches: APPROPO (Miryoosefi et al., 2019) and RCPO (Tessler et al., 2019). We note that none of the previous approaches in this setting address sample-efficient exploration. In addition, most of them are limited to linear constraints, with the exception of APPROPO (Miryoosefi et al., 2019), which can handle general convex constraints.[9] Both APPROPO and RCPO (used as a baseline by Miryoosefi et al., 2019) maintain and update a weight vector $\boldsymbol{\lambda}$, used to derive reward for an unconstrained RL algorithm, which we instantiate as A2C. APPROPO focuses on the feasibility problem, so it requires to specify a lower bound on the reward, which we set to $0.3$ for Mars rover and $0.1$ for Box. In the first two columns of Figure 1 we see that both versions of CONRL are able to solve the constrained RL task with a much smaller number of trajectories (see top two rows), and their overall consumption levels are substantially lower (the final row) than those of the previous approaches.

**Single-episode setting.** Closest to our work is TFW-UCRL2 (Cheung, 2019), which is based on UCRL (Jaksch et al., 2010). However, that approach focuses on the single-episode setting and requires a strong reachability assumption. By connecting terminal states of our MDP to the intial state, we reduce our episodic setting to single-episode setting in which we can compare CONRL against TFW-UCRL2. Results for Mars rover are depicted in last column of Figure 1.[10] Again, both versions of CONRL find the solution with a much smaller number of trajectories (note the log scale on the $x$-axis) and their overall consumption levels are much lower than those of TFW-UCRL2. This suggests that TFW-UCRL2 might be impractical in (at least some) episodic settings.

## 7 Conclusions

In this paper we study two types of constraints in the framework of constrained tabular episodic reinforcement learning: concave rewards and convex constraints, and knapsacks constraints. Our algorithms achieve near-optimal regret in both settings, and experimentally we show that our approach outperforms prior works on constrained reinforcement learning.

Regarding future work, it would be interesting to extend our framework to continuous state and action spaces. Potential directions include extensions to Lipschitz MDPs (Song and Sun, 2019) and MDPs with linear parameterization (Jin et al., 2019) where optimism-based exploration algorithms exist under the classic reinforcement learning setting without constraints.

## Broader Impact

Our work focuses on the theoretical foundations of reinforcement learning by addressing the important challenge of constrained optimization in reinforcement learning. We strongly believe that understanding the theoretical underpinnings of the main machine learning paradigms is essential and can guide principled and effective deployment of such methods.

Beyond its theoretical contribution, our work may help the design of reinforcement learning algorithms that go beyond classical digital applications of RL (board games and video games) and extend to settings with complex and often competing objectives. We believe that constraints constitute a fundamental limitation in extending RL beyond the digital world, as they exist in a wide variety of sequential decision-making applications (robotics, medical treatment, education, advertising). Our work provides a paradigm to design algorithms with efficient exploration despite the presence of constraints.

That said, one needs to ensure that an algorithm offers acceptable quality in applications. Any exploration method that does not rely on off-policy samples will inevitably violate constraints *sometimes* in order to learn. In some applications, this is totally acceptable: a car staying out of fuel in rare circumstances is not detrimental, an advertiser exhausting their budget some month is even less significant, a student dissatisfaction in an online test is unpleasant but probably acceptable. On the other hand, if the constraint violation involves critical issues like drug recommendation for severe diseases or decisions by self-driving cars that can cause physical harm to passengers then the algorithm needs to be carefully reviewed. It may be necessary to "prime" the algorithm with some data collected in advance (however costly it may be). One may need to make a judgement call on whether the ethical or societal standards are consistent with deploying an algorithm in a particular setting.

To summarize, our work is theoretical in nature and makes significant progress on a problem at the heart of RL. It has the potential to guide deployment of constrained RL methods in many important applications and tackle a fundamental bottleneck in deploying RL beyond the digital world. However, an application needs to be carefully reviewed before deployment.

## Acknowledgments and Disclosure of Funding

The authors would like to thank Rob Schapire for useful discussions that helped in the initial stages of this work. Part of the work was done when WS was at Microsoft Research NYC.

## Footnotes

[1] A fixed and known initial state is without loss of generality. In general, there is a fixed but unknown distribution $\rho$ from which the initial state is drawn before each episode. We modify the MDP by adding a new state $s_0$ as initial state, such that the next state is sampled from $\rho$ for any action. Then $\rho$ is "included" within the transition probabilities. The extra state $s_0$ does not contribute any reward and does not consume any resources.

[2]The max operator in Eq. (4) is to avoid dividing by 0.

[3] We refer the reader to the related work (in Section 1) for discussion on concurrent and independent papers. Unlike our results, these papers do not extend to either concave-convex or knapsack settings.

[4]Under mild assumptions, this program can be solved in polynomial time similar to its bandit analogue of Lemma 4.3 in (Agrawal and Devanur, 2014). We note that in the basic setting, it reduces to just a linear program.

[5]Note that in multi-armed bandit concave-convex setting (Agrawal and Devanur, 2014), proving feasibility of the best arm is straightforward as there are no transitions.

[6] Again, this is not an issue in multi-armed bandits.

[7]Code is available at `https://github.com/miryoosefi/ConRL`

[8]We are not aware of any benchmarks for convex/knapsack constraints. For transparency, we compare against prior works handling concave-convex or knapsack settings on established benchmarks for the linear case.

[9]In addition to that, trust region methods like CPO (Achiam et al., 2017) address a more restrictive setting and require constraint satisfaction at each iteration; for this reason, they are not included in the experiments.

[10]Due to a larger state space, it was computationally infeasible to run TFW-UCRL2 in the Box environment.

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
