[Supplementary Material]

## Structure of the supplementary material.

The supplementary material consists of six sections:

- Appendix A provides the formal description of the algorithm and the instantiations of CONPLANNER as well as how it can be expressed as a (linear/convex) mathematical program.

- Appendix B provides the proofs for the results of the basic setting presented in Section 3.

- Appendix C provides the proofs and additional discussion for the results of the concave-convex setting presented in Section 4.

- Appendix D provides the proofs and additional discussion for the results of knapsack setting presented in Section 5.

- Appendix E provides further details regarding the experiments presented in Section 6.

- Appendix F provides auxiliary concentration lemmas useful for the derivation of our results.

.

## A    Algorithm: Formal description and design choices

Our main algorithm, denoted by CONRL, is presented at Algorithm 1. We instantiate CONRL for our different settings (i.e. basic setting, concave-convex, and knapsack) by using the appropriate CONPLANNER that we discuss in the remainder of this section.

---

**Algorithm 1** CONRL

1: **for** Episode $k$ from 1 to $K$ **do**
2:    **Compute empirical estimates:**
                  Compute $N_k$, $\widehat{p}_k$, $\widehat{r}_k$, and $\widehat{c}_k$ based on equations (4-7)
3:    **Compute bonus:**
                  Compute $\widehat{b}_k$ as equation (9)
4:    **Call constrained planner:**
               $\pi_k \leftarrow$ CONPLANNER$(\widehat{p}_k, \widehat{r}_k, \widehat{c}_k, \widehat{b}_k)$
5:    **Execute policy:** initial state $s_{k,1} = s_0$
6:    **for** Stage $h$ from 1 to $H$ **do**
7:       Select $a_{k,h} \sim \pi_k\big(s_{k,h}\big)$
8:       Observe reward $r_{k,h}$, consumptions $\forall i \in \mathcal{D} : c_{k,h,i}$, and new state $s_{k,h+1}$
9:    **end for**
10: **end for**

---

### A.1    Basic setting - BASICCONPLANNER

We define the bonus-enhanced cMDP, i.e. $\mathcal{M}^{(k)} = \big(p^{(k)}, r^{(k)}, \boldsymbol{c}^{(k)}\big)$, as

$$p^{(k)}(s'|s,a) = \widehat{p}_k(s'|s,a) \quad \forall s,a,s'$$
$$r^{(k)}(s,a) = \widehat{r}_k(s,a) + \widehat{b}_k(s,a) \quad \forall s,a$$
$$c^{(k)}(s,a,i) = \widehat{c}_k(s,a,i) - \widehat{b}_k(s,a) \quad \forall s,a,i \in \mathcal{D}$$

then we solve the following optimization problem

$$\max_{\pi} \mathbb{E}^{\pi,p^{(k)}}\left[\sum_{h=1}^{H} r^{(k)}\big(s_h, a_h\big)\right] \quad \text{s.t.} \quad \forall i \in \mathcal{D} : \mathbb{E}^{\pi,p^{(k)}}\left[\sum_{h=1}^{H} c^{(k)}\big(s_h, a_h, i\big)\right] \leq \xi(i).$$

This optimization problem can be solved exactly since it is equivalent to the following linear program on occupation measures (Rosenberg and Mansour, 2019; Altman, 1999). Decision variables are

$\rho(s,a,h)$, i.e. probability of agent being at state action pair $(s,a)$ at time step $h$.

$$\max_{\rho} \sum_{s,a,h} \rho(s,a,h) r^{(k)}(s,a) \quad \text{s.t.} \quad \sum_{s,a,h} \rho(s,a,h) c^{(k)}(s,a,i) \leq \xi(i) \quad \forall i \in \mathcal{D}$$

$$\forall s',h \quad \sum_{a} \rho(s',a,h+1) = \sum_{s,a} \rho(s,a,h) p^{(k)}(s'|s,a) \quad (14)$$

$$\forall s,a,h \quad 0 \leq \rho(s,a,h) \leq 1 \quad \sum_{s,a} \rho(s,a,h) = 1$$

## A.2 Concave-convex setting - CONVEXCONPLANNER

In this setting, unlike basic setting, objective and constraints are not linear. Therefore, due to lack of monotonicity, we cannot explicitly define the bonus-enhanced cMDP $\mathcal{M}^{(k)} = \left(p^{(k)}, r^{(k)}, \boldsymbol{c}^{(k)}\right)$. The bonus-enhanced cMDP is implicit in the following program that we solve (see section 4)

$$\max_{\pi} \max_{r^{(k)} \in \left[\widehat{r}_k \pm \widehat{b}_k\right]} f\left(\mathbb{E}^{\pi,p^{(k)}}\left[\sum_{h=1}^{H} r^{(k)}\left(s_h,a_h\right)\right]\right) \text{ s.t.} \min_{\boldsymbol{c}^{(k)} \in \left[\widehat{\boldsymbol{c}}_k \pm \widehat{b}_k \cdot \mathbf{1}\right]} g\left(\mathbb{E}^{\pi,p^{(k)}}\left[\sum_{h=1}^{H} \boldsymbol{c}^{(k)}\left(s_h,a_h\right)\right]\right) \leq 0.$$

Similar to before, expressing this program based on occupation measures provides a convex program.

$$\max_{\rho} \max_{r \in \left[\widehat{r}_k \pm \widehat{b}_k\right]} f\left(\sum_{s,a,h} \rho(s,a,h) r(s,a)\right) \quad \text{s.t.} \min_{\boldsymbol{c} \in \left[\widehat{\boldsymbol{c}}_k \pm \widehat{b}_k \cdot \mathbf{1}\right]} g\left(\sum_{s,a,h} \rho(s,a,h) \boldsymbol{c}(s,a)\right) \leq 0$$

$$\forall s',h: \quad \sum_{a} \rho(s',a,h+1) = \sum_{s,a} \rho(s,a,h) \widehat{p}_k(s'|s,a)$$

$$\forall s,a,h: \quad 0 \leq \rho(s,a,h) \leq 1 \quad \text{and} \quad \sum_{s,a} \rho(s,a,h) = 1$$

$$(15)$$

The notations $r \in \left[\widehat{r}_k \pm \widehat{b}_k\right]$ and $\boldsymbol{c} \in \left[\widehat{\boldsymbol{c}}_k \pm \widehat{b}_k \cdot \mathbf{1}\right]$ are defined as

$$r \in \left[\widehat{r}_k \pm \widehat{b}_k\right] \iff \forall s,a: \quad r(s,a) \in [\widehat{r}_k(s,a) - \widehat{b}_k(s,a), \widehat{r}_k(s,a) + \widehat{b}_k(s,a)]$$

$$\boldsymbol{c} \in \left[\widehat{\boldsymbol{c}}_k \pm \widehat{b}_k \cdot \mathbf{1}\right] \iff \forall i \in \mathcal{D}, s,a: \quad c(s,a,i) \in [\widehat{c}_k(s,a,i) - \widehat{b}_k(s,a), \widehat{c}_k(s,a,i) + \widehat{b}_k(s,a)]$$

Note that if $f$ and $g$ are linear, we end up with a linear program similar to (14)

## A.3 Knapsack setting - KNAPSACKCONPLANNER

We define the bonus-enhanced cMDP, i.e. $\mathcal{M}^{(k)} = \left(p^{(k)}, r^{(k)}, \boldsymbol{c}^{(k)}\right)$ similar to basic setting (A.1). We also solve a similar optimization problem with tighter constraints:

$$\max_{\pi} \mathbb{E}^{\pi,p^{(k)}}\left[\sum_{h=1}^{H} r^{(k)}\left(s_h,a_h\right)\right] \quad \text{s.t.} \quad \forall i \in \mathcal{D}: \mathbb{E}^{\pi,p^{(k)}}\left[\sum_{h=1}^{H} c^{(k)}\left(s_h,a_h,i\right)\right] \leq \frac{(1-\epsilon)B_i}{K}.$$

This optimization problem can again be solved using the following linear program on occupation measures. Decision variables are $\rho(s,a,h)$, i.e. probability of agent being at state action pair $(s,a)$ at step $h$.

$$\max_{\rho} \sum_{s,a,h} \rho(s,a,h) r^{(k)}(s,a) \quad \text{s.t.} \quad \sum_{s,a,h} \rho(s,a,h) c^{(k)}(s,a,i) \leq \frac{(1-\epsilon)B_i}{K} \quad \forall i \in \mathcal{D}$$

$$\forall s',h \quad \sum_{a} \rho(s',a,h+1) = \sum_{s,a} \rho(s,a,h) p^{(k)}(s'|s,a) \quad (16)$$

$$\forall s,a,h \quad 0 \leq \rho(s,a,h) \leq 1 \quad \sum_{s,a} \rho(s,a,h) = 1$$

# B  Analysis: Basic setting (Section 3)

In this section, we prove the main guarantee for the basic setting.

## B.1 Validity of bonus (Lemma 3.2)

We first prove that $\widehat{b}_k(s,a) = H\sqrt{\frac{2\ln\left(8SAH(d+1)k^2/\delta\right)}{N_k(s,a)}}$ of Eq. (9) is valid as in the Definition 3.1.

*Proof of Lemma 3.2.* We focus on a single state-action pair $s, a$, stage $h$, and objective $m$. Since the support of $m$ is in $[0,1]$ and the one of the value is in $[0, H-1]$, by Hoeffding inequality (see Lemma F.2), it holds that, for all $k$, since $(s,a)$-pair is visited $N_k(s,a)$ times prior to episode $k$, with probability at least $1 - \delta'$:

$$\left|\left(\widehat{m}_k(s,a) - m^\star(s,a)\right) + \sum_{s'\in\mathcal{S}}\left(\widehat{p}_k(s'|s,a) - p^\star(s'|s,a)\right)V\right| \leq H\sqrt{\frac{2\ln(2/\delta')}{N_k(s,a)}}.$$

As a result, the bonus $\widehat{b}_k(s,a,\delta)$ satisfies this inequality for a particular state-action-step-objective with failure probability at most $\delta' = \frac{\delta}{4SAH(d+1)k^2}$ and is therefore valid (satisfying it for all states-actions-steps-objectives) with failure probability $\frac{\delta}{4k^2}$. Union bounding across episodes, the probability of $\widehat{b}_k(s,a,\delta)$ not being valid for some $k$ is at most $\sum_{k=1}^{K}\frac{\delta}{4k^2} \leq \delta$. $\qquad\square$

## B.2 Valid bonus implies optimism

The main reason to optimize a bonus-enhanced model with valid bonuses is because the latter render the model *optimistic*, i.e., its estimated reward is an overestimate of the true reward. Similarly, in constrained settings, its estimated resource consumptions are underestimates of the true resource consumptions. This is formalized in the following definition.

**Definition B.1.** A CMDP $\mathcal{M} = (p, r, \mathbf{c})$ is *optimistic* if its estimated reward (resp. consumption) value function for policy $\pi^\star$ upper (resp. lower) bounds its corresponding value function under the ground truth:

$$\mathbb{E}\left[V_r^{\pi^\star,p}(s_1,1)\right] \geq \mathbb{E}\left[V_{r^\star}^{\pi^\star,p^\star}(s_1,1)\right] \quad \text{and} \quad \mathbb{E}\left[V_{c_i}^{\pi^\star,p}(s_1,1)\right] \leq \mathbb{E}\left[V_{c_i^\star}^{\pi^\star,p^\star}(s_1,1)\right]\forall i \in \mathcal{D}.$$

An important block of the analysis for the basic setting is to show that, when using a bonus-enhanced model with valid bonuses, the resulting CMDP is optimistic.

**Lemma B.2.** *If the bonus $\widehat{b}_k(s,a)$ of Eq. (9) in episode $k$ is valid (Definition 3.1) for the corresponding CMDP $\mathcal{M}^{(k)} = \left(p^{(k)}, r^{(k)}, \mathbf{c}^{(k)}\right)$ then $\mathcal{M}^{(k)}$ is optimistic.*

*Proof.* We first prove the optimism of the model for the reward objective. More concretely, we show by induction that for any state $s$, action $a$, and stage $h$, $Q_{r^{(k)}}^{\pi^\star,p^{(k)}}(s,a,h) \geq Q_{r^\star}^{\pi^\star,p^\star}(s,a,h)$; taking expectation on the state-action pair of the first state, the claim then follows.

Since the setting ends at episode $H$, $Q_{r^{(k)}}^{\pi^\star,p^{(k)}}(s,a,H+1) = Q_{r^\star}^{\pi^\star,p^\star}(s,a,H+1) = 0$.

We assume that the inductive hypothesis $Q_{r^{(k)}}^{\pi^\star,p^{(k)}}(s,a,h+1) \geq Q_{r^\star}^{\pi^\star,p^\star}(s,a,h+1)$ (and thus also $V_{r^{(k)}}^{\pi^\star,p^{(k)}}(s,h+1) \geq V_{r^\star}^{\pi^\star,p^\star}(s,h+1)$) holds, and proceed with the inductive step. The $Q$-functions in question are:

$$Q_{r^{(k)}}^{\pi^\star,p^{(k)}}(s,a,h) = r^{(k)}(s,a) + \sum_{s'\in\mathcal{S}}p^{(k)}(s'|s,a)V_{r^{(k)}}^{\pi^\star,p^{(k)}}(s',h+1)$$

$$\geq r^{(k)}(s,a) + \sum_{s'\in\mathcal{S}}p^{(k)}(s'|s,a)V_{r^\star}^{\pi^\star,p^\star}(s',h+1)$$

$$Q_{r^\star}^{\pi^\star,p^\star}(s,a,h) = r^\star(s,a) + \sum_{s'\in\mathcal{S}}p^\star(s'|s,a)V_{r^\star}^{\pi^\star,p^\star}(s',h+1)$$

Subtracting, we have:

$$Q_{r^{(k)}}^{\pi^\star,p^{(k)}}(s,a,h) - Q_{r^\star}^{\pi^\star,p^\star}(s,a,h) \geq \left(\widehat{r}_k(s,a) + \widehat{b}_k(s,a) - r^\star(s,a)\right)$$

$$+ \sum_{s'\in\mathcal{S}}\left(\widehat{p}_k(s'|s,a) - p^\star(s'|s,a)\right)V_{r^\star}^{\pi^\star,p^\star}(s',h+1) \geq 0,$$

where the last inequality holds since the bonuses are valid.

The optimism of the model with respect to the consumption objectives follows the same steps altering the direction of the inequalities and setting the estimate as empirical mean minus the bonus. □

We emphasize that our bonus in Eq (9) does not scale polynomially with respect to $|\mathcal{S}|$; despite that, as indicated by the above lemma, it suffices to prove optimism.

### B.3 Simulation lemma

To prove the Bellman-error regret decomposition, an essential piece is the so called *simulation lemma* (Kearns and Singh, 2002) which we adapt to constrained settings below:

**Lemma B.3** (Simulation lemma)**.** *For any policy $\pi$, any* CMDP *$\mathcal{M} = (p, r, \boldsymbol{c})$, and any objective $m \in \{r\} \cup \{c_i\}_{i \in \mathcal{D}}$ with corresponding true objective $m^\star \in \{r^\star\} \cup \{c_i^\star\}_{i \in \mathcal{D}}$,, it holds that:*

$$\mathbb{E}^\pi\Big[V_m^{\pi,p}(s_1, 1)\Big] - \mathbb{E}^\pi\Big[V_{m^\star}^{\pi,p^\star}(s_1, 1)\Big] = \mathbb{E}^\pi\Big[\sum_{h=1}^{H} \text{BELL}_m^{\pi,p}(s_h, a_h, h)\Big]. \tag{17}$$

*Proof.* For all of $m \in \{r\} \cup \{c_i\}_{i \in \mathcal{D}}$, rearranging the definitions of Bellman errors, we obtain:

$$Q_m^{\pi,p}(s, a, h) = \Big(\text{BELL}_m^{\pi,p}(s, a, h) + m^\star(s, a)\Big) + \sum_{s' \in \mathcal{S}} p^\star(s'|s, a) V_m^{\pi,p}(s', h+1)$$

$$Q_{m^\star}^{\pi,p^\star}(s, a, h) = \Big(\text{BELL}_{m^\star}^{\pi,p^\star}(s, a, h) + m^\star(s, a)\Big) + \sum_{s' \in \mathcal{S}} p^\star(s'|s, a) V_{m^\star}^{\pi,p^\star}(s', h+1)$$

By definition of the Bellman error, the Bellman error with respect to the true model is equal to 0. As a result, subtracting the two above equations, we obtain:

$$Q_m^{\pi,p}(s, a, h) - Q_{m^\star}^{\pi,p^\star}(s, a, h) = \text{BELL}_m^{\pi,p}(s, a, h) + \sum_{s' \in \mathcal{S}} p^\star(s'|s, a)\Big(V_m^{\pi,p}(s', h+1) - V_{m^\star}^{\pi,p^\star}(s', h+1)\Big).$$

Taking expectation over policy $\pi$ to select $a$, the initial state $s_1$, and setting $h = 1$, we obtain:

$$\mathbb{E}_{s_1}\Big[V_m^{\pi,p}\big(s(1), 1\big) - V_{m^\star}^{\pi,p^\star}(s_1, 1)\Big] = \mathbb{E}^\pi\Big[\text{BELL}_m^{\pi,p}\big(s_1, a_1, 1\big)\Big] + \mathbb{E}^\pi\Big[V_m^{\pi,p}\big(s_2, 2\big) - V_{m^\star}^{\pi,p^\star}\big(s_2, 2\big)\Big].$$

Recursively bounding the second term of the RHS as above concludes the lemma. □

### B.4 Bellman-error regret decomposition (Proposition 3.3)

*Proof of Proposition 3.3.* The consumption requirement (11) for resource $i$ follows by applying the simulation lemma (Lemma B.3) on CMDP $\mathcal{M}^{(k)}$ and objective $m = c_i^{(k)}$ (with corresponding true objective $m^\star = c_i^\star$) and using that $\pi_k$ is feasible for CONPLANNER$(p^{(k)}, r^{(k)}, \boldsymbol{c}^{(k)})$:

$$\mathbb{E}^{\pi_k, p^\star}\Big[\sum_{h=1}^{H} c^\star(s_h, a_h, i)\Big] = \mathbb{E}^{\pi_k}\Big[V_{c_i^\star}^{\pi,p^\star}(s_1, 1)\Big] = \mathbb{E}\Big[V_{c_i}^{\pi_k,p}(s_1, 1)\Big] - \mathbb{E}^{\pi_k}\Big[\sum_{h=1}^{H} \text{BELL}_{c_i^{(k)}}^{\pi_k,p^{(k)}}\big(s_h, a_h, h\big)\Big]$$

$$\leq \xi(i) + \mathbb{E}^{\pi_k}\Big[\sum_{h=1}^{H} \Big|\text{BELL}_{c_i^{(k)}}^{\pi_k,p^{(k)}}\big(s_h, a_h, h\big)\Big|\Big]$$

Regarding the reward requirement (10), what we wish to bound is:

$$\mathbb{E}^{\pi^\star, p^\star}\Big[\sum_{h=1}^{H} r^\star(s_h, a_h)\Big] - \mathbb{E}^{\pi_k, p^\star}\Big[\sum_{h=1}^{H} r^\star(s_h, a_h)\Big] = \mathbb{E}\Big[V_{r^\star}^{\pi^\star,p^\star}(s_1, 1)\Big] - \mathbb{E}\Big[V_{r^\star}^{\pi_k,p^\star}(s_1, 1)\Big]$$

the validity of the bonus implies that the model $\mathcal{M}^{(k)}$ is optimistic (Lemma B.2), i.e., we have that $\mathbb{E}\Big[V_{r^\star}^{\pi^\star,p^\star}(s_1, 1)\Big] \leq \mathbb{E}\Big[V_{r^{(k)}}^{\pi^\star,p^{(k)}}(s_1, 1)\Big]$. If $\pi^\star$ is feasible for CONPLANNER$(p^{(k)}, r^{(k)}, \boldsymbol{c}^{(k)})$ then, since $\pi_k$ is the maximizer for this program:

$$\mathbb{E}\Big[V_{r^{(k)}}^{\pi^\star,p^{(k)}}(s_1, 1)\Big] - \mathbb{E}\Big[V_{r^\star}^{\pi_k,p^\star}(s_1, 1)\Big] \leq \mathbb{E}\Big[V_{r^{(k)}}^{\pi_k,p^{(k)}}(s_1, 1)\Big] - \mathbb{E}\Big[V_{r^\star}^{\pi_k,p^\star}(s_1, 1)\Big] \tag{18}$$

$$= \mathbb{E}^{\pi_k}\Big[\sum_{h=1}^{H} \text{BELL}_{r^{(k)}}^{\pi_k,p^{(k)}}\big(s_h, a_h, h\big)\Big]$$

where the last equality holds by applying the simulation lemma with $m = r$. Hence, this proves (10).

What is left to show is that $\pi^\star$ is indeed feasible for $\text{CONPLANNER}(p^{(k)}, r^{(k)}, \boldsymbol{c}^{(k)})$. Since $\mathcal{M}^{(k)}$ is optimistic and $\pi^\star$ is feasible for the ground truth $\mathcal{M}^\star$, for all resources $i \in \mathcal{D}$:

$$\mathbb{E}\left[V_{c_i^{(k)}}^{\pi^\star, p^{(k)}}(s_1, 1)\right] \leq \mathbb{E}\left[V_{c_i^\star}^{\pi^\star, p^\star}(s_1, 1)\right] \leq \xi(i).$$

This completes the proof of the proposition. $\qquad\square$

### B.5 Bounding the Bellman error

We now provide an upper bound on the Bellman error which arises in the RHS of the regret decomposition (Proposition 3.3).

**Lemma B.4.** *Let $\epsilon > 0$. If the bonus $\widehat{b}_k$ is valid for all episodes $k$ simultaneously then, with probability at least $1 - \delta$: for all objectives $m^{(k)} \in \{r^{(k)}\} \cup \{c_i^{(k)}\}_{i \in \mathcal{D}}$, transitions $p = p^{(k)}$, and stages $h$, the Bellman error at episode $k$ is upper bounded by:*

$$\left|\text{BELL}_{m^{(k)}}^{\pi_k, p^{(k)}}(s, a, h)\right| \leq 2H\sqrt{\frac{2S\ln\left(16SAH^2(d+1)k^2/(\epsilon\delta)\right)}{N_k(s, a)}} + \epsilon S.$$

*Proof of Lemma B.4.* Let $\Psi$ be an $\epsilon$-net in $[-(H-1), (H-1)]^S$. For a fixed value $\bar{V} \in \Psi$, similar to Lemma 3.2, with probability $1 - \delta'$, simultaneously for all states $s \in \mathcal{S}$, actions $a \in \mathcal{A}$, steps $h \in [H]$, episodes $k \in [K]$, and objectives $m^{(k)} \in \{r^{(k)}\} \cup \{c_i^{(k)}\}_{i \in \mathcal{D}}$, it holds that:

$$\left|m^{(k)}(s, a) - m^\star(s, a) + \sum_{s' \in \mathcal{S}}\left(p(s'|s, a) - p^\star(s'|s, a)\right)\bar{V}(s')\right|$$

$$\leq \widehat{b}_k(s, a) + H\sqrt{\frac{2\ln\left(8SAH(d+1)k^2/\delta'\right)}{N_k(s, a)}}$$

Since $\Psi$ is an $\epsilon$-net for $\bar{V}$, there are $(2H/\epsilon)^S$ potential values. In order to have the above hold simultaneously for all these values with probability $1 - \delta$, we need to set $\delta' = \frac{\delta}{(2H/\epsilon)^S}$.

Since the value $\left(p^{(k)}(s'|s, a) - p^\star(s'|s, a)\right)V_{m^{(k)}}^{\pi_k, p}(s', h+1)$ is in $[-(H-1), (H-1)]$ for all $s'$, it holds that there exists a value $V$ in the $\epsilon$-net with distance at most $\epsilon S$. As a result, since $\widehat{b}_k(s, a)$ is valid for $k$:

$$\left|\text{BELL}_{m^{(k)}}^{\pi_k, p^{(k)}}(s, a, h)\right| \leq \left|m^{(k)}(s, a) - m^\star(s, a) + \sum_{s' \in \mathcal{S}}\left(p^{(k)}(s'|s, a) - p^\star(s'|s, a)\right)V(s')\right|$$

$$+ \left|\sum_{s' \in \mathcal{S}}\left(p^{(k)}(s'|s, a) - p^\star(s'|s, a)\right)\left(V(s') - V_{m^{(k)}}^{\pi_k, p^{(k)}}(s', h+1)\right)\right|$$

$$\leq \widehat{b}_k(s, a) + H\sqrt{\frac{2S\ln\left(16SAH^2(d+1)k^2/(\epsilon\delta)\right)}{N_k(s, a)}} + \epsilon S.$$

Upper bounding $\widehat{b}_k(s, a) \leq H\sqrt{\frac{2S\ln\left(16SAH^2(d+1)k^2/(\epsilon\delta)\right)}{N_k(s, a)}}$ completes the lemma. $\qquad\square$

### B.6 Final guaraantee for the basic setting (Theorem 3.4)

*Proof.* The failure probability of the algorithm is $\delta$ due to the validity of bonus $\widehat{b}_k(s, a)$ (Lemma 3.2) and another $\delta$ by the bound on Bellman error (Lemma B.4). When neither failure events occur (probability $1 - 2\delta$), Proposition 3.3 upper bounds either of reward or consumption regret by $\mathbb{E}^{\pi_k}\left[\left|\text{BELL}_{m^{(k)}}^{\pi_k, p^{(k)}}(s_h, a_h, h)\right|\right]$. By Lemma B.4, the Bellman error at episode $t$, for $\epsilon > 0$, is at most:

$$\left|\text{BELL}_{m^{(t)}}^{\pi_t, p^{(t)}}(s_{t,h}, a_{t,h}, h)\right| \leq 2H\sqrt{\frac{2S\ln\left(16SAH^2(d+1)t^2/(\epsilon\delta)\right)}{N_t(s, a)}} + \epsilon S$$

Summing across all $h = 1 \ldots H$ and $t = 1, \ldots, k$, the sum of Bellman errors is at most:

$$\sum_{t=1}^{k} \sum_{h=1}^{H} \left| \text{BELL}_{m^{(t)}}^{\pi_t, p^{(t)}}(s_{t,h}, a_{t,h}, h) \right|$$

$$\leq \sum_{t=1}^{k} \sum_{h=1}^{H} \left( 2H \sqrt{\frac{2S \ln \left( 16SAH^2(d+1)t^2/(\epsilon\delta) \right)}{N_t(s,a)}} + \epsilon S \right)$$

$$\leq \sum_{s,a} \left( \sum_{j=1}^{2H} 2H \sqrt{2S \ln \left( 16SAH^2(d+1)k^2/(\epsilon\delta) \right)} \right.$$

$$\left. + \sum_{j=H+1}^{N_k(s,a)} 2H \sqrt{\frac{4S \ln \left( 16SAH^2(d+1)k^2/(\epsilon\delta) \right)}{j}} + \epsilon S \right)$$

The second inequality follows since a particular state-action pair may have the same visitations for $H$ times (as we only update this quantity at the end of the episode). To avoid incurring an additional dependence on $H$, we separate the first $H$ visitations of each state-action pair and treat the bound as if $j = 1$ for them. [11] For the remaining visitations, $j$ and $N_k(s,a)$ are always within a factor of 2 and this factor therefore appears within the square root.

We now bound the second term:

$$\sum_{s,a} \left( \sum_{j=H+1}^{N_k(s,a)} 2H \sqrt{\frac{4S \ln \left( 16SAH^2(d+1)k^2/(\epsilon\delta) \right)}{j}} + \epsilon S \right)$$

$$\leq 2SAH \sqrt{N_k(s,a) \ln \left( N_k(s,a) \right) \cdot 4S \ln \left( 16SAH^2(d+1)k^2/(\epsilon\delta) \right)} + \epsilon k H S$$

$$\leq 2SAH \sqrt{\frac{kH \cdot 4S \cdot \ln(k) \ln \left( 16SAH^2(d+1)k^2/(\epsilon\delta) \right)}{SA}} + \epsilon k H S$$

$$\leq 8S\sqrt{AH^3} \cdot \sqrt{k} \cdot \sqrt{\ln(k) \ln \left( 2SAH(d+1)k/\delta \right)} + 1.$$

The last inequality holds by setting $\epsilon = \frac{1}{kHS}$.

The first term can be bounded by additive terms that depend only logarithmically on $k$:

$$\sum_{s,a} \left( \sum_{j=1}^{2H} 2H \sqrt{2S \ln \left( 16SAH^2(d+1)k^2/(\epsilon\delta) \right)} \leq 16S^{3/2} AH^2 \sqrt{\ln(2SAH(d+1)k/\delta)} \right.$$

As a result:

$$\sum_{t=1}^{k} \sum_{h=1}^{H} \left| \text{BELL}_{m^{(t)}}^{\pi_t, p^{(t)}}(s_{t,h}, a_{t,h}, h) \right| \leq 8S\sqrt{AH^3}\sqrt{k} \cdot \sqrt{\ln(k) \ln \left( 2SAH(d+1)k/\delta \right)} + 1$$

$$+ 16S^{3/2} AH^2 \sqrt{\ln \left( 2SAH(d+1)k/\delta \right)}$$

Now we link the additive Bellman error to the expected sum of Bellman errors under the expectation of the policies $\{\pi_t\}$ (as needed by Proposition 3.3) via a simple martingale argument. From Lemma F.3, with probability at least $1 - \delta$, we have:

$$\left| \sum_{t=1}^{k} \sum_{h=1}^{H} \left| \text{BELL}_{m^{(t)}}^{\pi_t, p^{(t)}}(s_{t,h}, a_{t,h}, h) \right| - \sum_{t=1}^{k} \sum_{h=1}^{H} \mathbb{E}^{\pi_t} \left[ \sum_{h=1}^{H} \left| \text{BELL}_{m^{(t)}}^{\pi_t, p^{(t)}}(s_h, a_h, h) \right| \right] \right|$$

$$\leq H^{1.5}\sqrt{2 \ln(4k^2/\delta)k},$$

where we use the fact that $|\text{BELL}^{\pi, p}| \leq 3H$ due to of $Q_m^{\pi, p}(s, a) \in [0, H]$, $m^\star(s, a) \in [0, 1]$, and $V_m^{\pi, p}(s) \in [0, H]$. Combining the above, we conclude the proof. □

# C Analysis: concave-convex setting (Section 4)

In this section, we prove the main guarantee for the convex-concave setting. Since the regret decomposition of the basic setting (Proposition 3.3) does not hold direclty as $f$ and $g$ are not linear, we need to create an analogous regret decomposition (Proposition C.2) for the convex-concave setting. This can be done by leveraging the Lipschitzness of the functions. Armed with this new regret decomposition, we can directly call the results we have for for the basic setting (e.g., upper bounds of Bellman errors) to conclude the regret analysis for the convex-concave setting. The first step leading to this regret decomposition is to show that $\pi^\star$ is a feasible solution of CONVEXCONPLANNER.

## C.1 Feasibility of optimal policy in concave-convex setting (Lemma C.1)

**Lemma C.1.** *If the bonus $\widehat{b}_k$ is valid (in the sense of Definition 3.1) then policy $\pi^\star$ that maximizes the objective of the convex-concave setting is feasible in CONVEXCONPLANNER.*

*Proof.* Unlike the linear case, the feasibility of $\pi^\star$, requires more care. Applying the same dynamic programming arguments as in Lemma B.2, it follows that:

$$\forall i \in \mathcal{D}: \qquad \mathbb{E}\left[V^{\pi^\star, p^{(k)}}_{\widehat{c}_{i,k} - b_k}(s_1, 1)\right] \leq \mathbb{E}_s\left[V^{\pi^\star, p^\star}_{c_i^\star}(s_1, 1)\right] \leq \mathbb{E}\left[V^{\pi^\star, p^{(k)}}_{\widehat{c}_{i,k} + b_k}(s_1, 1)\right].$$

Letting $\widetilde{g}(\alpha) = \mathbb{E}\left[V^{\pi^\star, p^{(k)}}_{\widehat{c}_{i,k} + \alpha b_k}(s(1), 1)\right]$, the above can be rewritten as:

$$\forall i \in \mathcal{D}: \qquad \widetilde{g}(-1) \leq \mathbb{E}\left[V^{\pi^\star, p^\star}_{c_i^\star}(s_1, 1)\right] \leq \widetilde{g}(1).$$

Since $\widetilde{g}(\cdot)$ is the expected value over the same policy and under the same transitions, it is continuous with respect to its argument. As a result, applying mean-value theorem on each $i$ separately, there exists some $\alpha_i$ such that $\widetilde{g}(\alpha_i) = \mathbb{E}_s\left[V^{\pi^\star, p^\star}_{c_i^\star}(s_1, 1)\right]$. Due to the feasibility of $\pi^\star$ on the true transitions and consumptions, it holds that $g\left(\widetilde{\boldsymbol{g}}(\alpha_i)\right) \leq 0$. Hence, selecting estimates $\widehat{c}_{i,k} + \alpha_i \widehat{b}_k$ creates a feasible solution for $\pi^\star$ under the estimated transitions of the CONVEXCONPLANNER program. The final value of $\pi^\star$ at this program maximizes the objective retaining feasibility; hence the existence of one feasible selection of consumption estimates concludes the proof of the lemma. $\square$

We conclude by remarking that proving optimism feasibility for the concave-convex setting in multiple-step RL setting is more challenging than that in single-step multi-arm bandit setting Agrawal and Devanur (2014) since in bandits, there are no transitions. In the proof above, to show that $\pi^\star$ is feasible in CONVEXCONPLANNER which is defined with respect to $p^{(k)}$, we leverage the fact that $\widetilde{g}(\alpha)$ is continuous and a novel application of mean-value theorem to link $\pi^\star$'s performance in the optimistic model $\mathbb{E}\left[V^{\pi^\star, p^{(k)}}_{\widehat{c}_{i,k} + \alpha_i b_k}(s_1, 1)\right]$ and $\pi^\star$'s performance under the real model $\mathbb{E}_s\left[V^{\pi^\star, p^\star}_{c_i^\star}(s_1, 1)\right]$.

## C.2 Regret decomposition for concave-convex setting

Using the Lipschitz continuous assumption of $f$ and $g$, we can decompose the regret into a sum of Bellman errors as before, but scaled by the Lipschitz constant this time.

**Proposition C.2.** *Let $L$ be the Lipschitz constant for $f$ and $g$. If $\widehat{b}_k(s, a, \delta)$ is valid for all episodes $k$ simultaneously then the per-episode reward and consumption regrets can be upper bounded by:*

$$f\left(\mathbb{E}^{\pi^\star, p^\star}\left[\sum_{h=1}^H r^\star(s_h, a_h)\right]\right) - f\left(\mathbb{E}^{\pi_k, p^\star}\left[\sum_{h=1}^H r^\star(s_h, a_h)\right]\right) \leq L \cdot \mathbb{E}^{\pi_k}\left[\sum_{h=1}^H \text{BELL}^{\pi_k, p^{(k)}}_{r^{(k)}}(s_h, a_h, h)\right]$$

$$g\left(\mathbb{E}^{\pi_k, p^\star}\left[\sum_{h=1}^H \boldsymbol{c}^\star(s_h, a_h, i)\right]\right) \leq L \sum_{i \in \mathcal{D}} \cdot \mathbb{E}^{\pi_k}\left[\sum_{h=1}^H \left|\text{BELL}^{\pi_k, p^{(k)}}_{c_i^{(k)}}(s_h, a_h, h)\right|\right]$$

*Proof.* We first prove the reward requirement. Let $r(\pi)$ be the solution of the inner maximization program for policy $\pi$, and we define $r^{(k)} = r(\pi_k)$. For notational convenience, we denote $V^{\pi, p}_m =$

$\mathbb{E}^{\pi,p}\left[V_m^{\pi,p}\right]$ Since $r^\star(s,a) \in [\widehat{r}(s,a) - \widehat{b}_k(s,a,\delta), \widehat{r}(s,a) + \widehat{b}_k(s,a,\delta)]$ and the bonus $\widehat{b}_k$ is valid, similar to Lemma B.2, it holds:

$$V_{r^\star}^{\pi^\star,p^\star} \in \left[V_{\widehat{r}-b}^{\pi^\star,p^{(k)}}, V_{\widehat{r}+b}^{\pi^\star,p^{(k)}}\right]. \tag{19}$$

As a result, by mean-value theorem, there exists $\alpha \in [-1,1]$ such that $V_{r^\star}^{\pi^\star,p^\star} = V_{\widehat{r}+\alpha b}^{\pi^\star,p^{(k)}}$. Since $\pi_k$ is the maximizer of CONVEXCONPLANNER and $\pi^\star$ is feasible for that program, it holds that:

$$f\left(V_{r(\pi_k)}^{\pi_k,p^{(k)}}\right) \ge f\left(V_{r(\pi^\star)}^{\pi^\star,p^{(k)}}\right) \ge f\left(V_{\widehat{r}+\alpha b}^{\pi^\star,p^{(k)}}\right) = f\left(V_{r^\star}^{\pi^\star,p^\star}\right), \tag{20}$$

where the second-to-last inequality holds since $r(\pi^\star)$ is the maximizer of the inner program for $\pi^\star$ and the equality holds by (19).

We are now ready to provide the equivalent of the regret decomposition:

$$f(V_{r^\star}^{\pi^\star,p^\star}) - f(V_{r^\star}^{\pi_k,p^\star}) \le f(V_{r(\pi_k)}^{\pi_k,p^{(k)}}) - f(V_{r^\star}^{\pi_k,p^\star}) \le L \cdot \left|V_{r(\pi_k)}^{\pi_k,p^{(k)}} - V_{r^\star}^{\pi_k,p^\star}\right|$$

$$\le L \cdot \mathbb{E}^{\pi_k}\left(\sum_{h=1}^{H} \mathrm{BELL}_{r^{(k)}}^{\pi_k,p^{(k)}}(s_h,a_h,h)\right)$$

where the first inequality holds by (20). the second inequality by Lipschitzness and the last inequality holds by simulation lemma (Lemma B.3).

For the consumption requirement, since $\pi_k$ is feasible in CONVEXCONPLANNER, denoting again by $c(\pi)$ the consumption in the maximizer for policy $\pi$ in the inner mathematical program. Same as above we define $c^{(k)} = c(\pi_k)$. It holds that:

$$g\left(\mathbb{E}^{\pi_k,p^{(k)}}\left[\sum_{h=1}^{H} c_h(\pi_k)\right]\right) \le 0 \tag{21}$$

As a result,

$$g\left(\mathbb{E}^{\pi_k,p^\star}\left[\sum_{h=1}^{H} c_h^\star\right]\right) - g\left(\mathbb{E}^{\pi_k,p^{(k)}}\left[\sum_{h=1}^{H} c_h(\pi_k)\right]\right) \le L \left\|\mathbb{E}^{\pi_k,p^\star}\left[\sum_{h=1}^{H} c_h^\star\right] - \mathbb{E}^{\pi_k,p^{(k)}}\left[\sum_{h=1}^{H} c_h(\pi_k)\right]\right\|_1$$

$$= L \sum_{i \in \mathcal{D}} \left|\mathbb{E}^{\pi_k,p^\star}\left[\sum_{h=1}^{H} c_h^\star(i)\right] - \mathbb{E}^{\pi_k,p^{(k)}}\left[\sum_{h=1}^{H} c_h(\pi_k,i)\right]\right|$$

$$\le L \cdot \sum_{i \in \mathcal{D}} \mathbb{E}^{\pi}\left(\sum_{h=1}^{H}\left|\mathrm{BELL}_{c_i^{(k)}}^{\pi_k,p^{(k)}}(s_h,a_h,h)\right|\right),$$

where again we applied Lipschitness and simulation lemma. $\qquad\square$

### C.3  Concave-convex theorem (Theorem 4.1)

*Proof of Theorem 4.1.* The proof follows similarly to the proof of Theorem 3.4 by replacing Proposition 3.3 with Proposition C.2. The linear dependency on $d$ in the consumption regret comes from the fact that the Lipschitzness of $g$ is defined in L1 norm. $\qquad\square$

## D  Analysis: Knapsack setting (Section 5)

In this section, we prove the guarantee for the hard-constraint setting. The goal is to show that over $K$ episodes, our algorithm has sublinear reward regret comparing to the best dynamic policy (formally defined in Appendix D.2), while satisfying hard budget constraints with high probability.

## D.1 Theorem with hard constraints (Theorem 5.1)

*Proof of Theorem 5.1.* We denote by OPT the expected total reward of $\pi^\star$. Consider now the policy $\widetilde{\pi}^\star$ that selects the null policy with probability $\epsilon$ and follows $\pi^\star$ otherwise. This policy is feasible for (13); as a result the expected reward $\widetilde{\pi}^\star$ for (13) is at least $(1-\epsilon)$OPT. Since the total reward is upper bounded by $KH$, it therefore holds that:

$$\sum_{k=1}^{K} \mathbb{E}^{\widetilde{\pi}^\star}\left[\sum_{h=1}^{H} r^\star(s_h, a_h)\right] \geq (1-\epsilon)\text{OPT} \geq \text{OPT} - \epsilon KH \tag{22}$$

In the high-probability event where the regret guarantee of $\text{AGGREG}(\delta)$ does not fail, the reward of the algorithm is at least:

$$\sum_{k=1}^{K}\sum_{h=1}^{H} r_{k,h} \geq \sum_{k=1}^{K} \mathbb{E}^{\widetilde{\pi}^\star}\left[\sum_{h=1}^{H} r^\star(s_h, a_h)\right] - \text{AGGREG}(\delta), \tag{23}$$

Combining (22) and (23), with probability $1-\delta$, the reward regret with respect to $\pi^\star$ is at most:

$$\text{REWREG(K)} \leq \frac{1}{K}\text{AGGREG}(\delta) + \epsilon H \tag{24}$$

We now focus on the consumption. Since we optimize (13), for any resource $i \in \mathcal{D}$, when the regret guarantee $\text{AGGREG}(\delta)$ against $\widetilde{\pi}^\star$ does not fail and given that $\widetilde{\pi}^\star$ is feasible for (13), it holds that:

$$\sum_{k=1}^{K}\sum_{h=1}^{H} c_{k,h,i} \leq \sum_{k=1}^{K} \mathbb{E}^{\widetilde{\pi}^\star}\left[\sum_{h=1}^{H} c(s_h, a_h, i)\right] + \text{AGGREG}(\delta) \leq (1-\epsilon)B_i + \text{AGGREG}(\delta)$$

Hence, when the regret guarantee $\text{AGGREG}(\delta)$ does not fail, the consumption is less than $B_i$ for all $i$ as long as $\epsilon \geq \frac{\text{AGGREG}(\delta)}{\min_i B_i}$. Moreover $\epsilon$ is a probability as a result it should also be less than 1 which holds when $\min_i B_i \geq \text{AGGREG}(\delta)$. Applying on (24) and assuming without loss of generality that $KH > \min_i B_i$ (otherwise the setting is essentially unconstrained), the reward regret is at most

$$\text{REWREG}(K) \leq \frac{2H\text{AGGREG}(\delta)}{\min_i B_i}.$$

$\square$

## D.2 Dynamic policy benchmark

We call a policy *dynamic* if it maps the entire history to a distribution over the action space. Specifically we denote history $\mathcal{H}_{k,h}$ as the history that contains all the information from the beginning of the first episode to the end of the step $h-1$ at the $k$-th episode plus the state at step $h$ in episode k. At any episode k and step $h$, a dynamic policy $\widetilde{\pi}(\cdot|\mathcal{H}_{k;h}) \in \Delta(\mathcal{A})$ maps history $\mathcal{H}_{k;h}$ to a distribution over action space. We denote $\Pi_{\text{dynamic}}$ as the set of all dynamic policies that satisfies the budget constraints deterministically, i.e., for any $\widetilde{\pi} \in \Pi_{\text{dynamic}}$, when executed for $K$ episodes in the MDP, we have $\sum_{k=1}^{K}\sum_{h=1}^{H} c_i(s_{k,h}, a_{k,h}) \leq B_i$ for all $i \in \mathcal{D}$, deterministically. Ideally we want to compare against the best dynamic policy that maximizes the expected total reward $\max_{\widetilde{\pi} \in \Pi_{\text{dynamic}}} \mathbb{E}^{\widetilde{\pi}}\left[\sum_{k=1}^{K}\sum_{h=1}^{K} r_{k,h}\right]$. We denote such an optimal dynamic policy as $\widetilde{\pi}^\star$ and its expected total reward across K episodes as

$$\text{OPT} := \max_{\widetilde{\pi} \in \Pi_{\text{dynamic}}} \mathbb{E}^{\widetilde{\pi}}\left[\sum_{k=1}^{K}\sum_{h=1}^{K} r_{k,h}\right].$$

The lemma below shows that indeed the stationary Markovian policy $\pi^\star$ actually achieves no smaller expected total reward across K episodes than that of the best dynamic policy.

**Lemma D.1.** *The reward of the policy $\pi^\star$ maximizing program (1) with $\xi(i) = \frac{B_i}{K}$ is at least as large as the per-episode reward of the optimal dynamic policy that is subject to hard constraints instead:*

$$\mathbb{E}^{\pi^\star}\left[\sum_{h=1}^{H} r^\star(s_h, a_h)\right] \geq \frac{1}{K}\max_{\widetilde{\pi} \in \Pi_{dynamic}} \mathbb{E}^{\widetilde{\pi}}\left[\sum_{k=1}^{K}\sum_{h=1}^{H} r(s_{k,h}, a_{k,h})\right] = \frac{\text{OPT}}{K}.$$

*Proof.* Denote $\widetilde{\pi}^\star$ as the optimal dynamic policy from $\Pi_{\text{dynamic}}$. Any policy induces a state-action distribution at episode $k$ and stage $h$, denoted as $\rho_{\widetilde{\pi}}(s, a; h, k)$, which stands for the probability of $\widetilde{\pi}$ visits state-action pair $(s, a)$ at stage $h$ in episode $k$. Denote $\rho_{\widetilde{\pi}}(s, a; h) = \sum_{k=1}^{K} \rho_{\widetilde{\pi}}(s, a; h, k)/K$ which stands for the probability of $\widetilde{\pi}$ visiting $(s, a)$ at stage $h$. We have:

$$\sum_a \rho_{\widetilde{\pi}}(s', a; h, k) = \sum_{s,a} \rho_{\widetilde{\pi}}(s, a; h - 1, k)p^\star(s'|s, a), \forall s',$$

due to the Markovian transition $p^\star(s'|s, a)$, which implies that:

$$\sum_a \rho_{\widetilde{\pi}}(s', a; h) = \sum_{s,a} \rho_{\widetilde{\pi}}(s, a; h - 1)p^\star(s'|s, a), \forall s'.$$

Hence, $\rho_{\widetilde{\pi}}(s, a; h)$ satisfies the flow constraints, and hence induces a stationary Markovian policy:

$$\pi_{\widetilde{\pi}}(a|s) \propto \rho_{\widetilde{\pi}}(s, a; h)/ \sum_a \rho_{\widetilde{\pi}}(s, a; h),$$

and $\pi_{\widetilde{\pi}}$ induces state-action visitation distribution that are exactly equal to $\rho_{\widetilde{\pi}}(s, a; h)$.

Note that $\widetilde{\pi}^\star$ satisfies the budget constraints deterministically, which means in expectation, it will satisfies the constraints as well, i.e.,

$$\sum_{k=1}^{K}\sum_{h=1}^{H}\sum_{(s,a)} \rho_{\widetilde{\pi}^\star}(s, a; h)c_i(s, a) \le B_i, \quad \forall i \in \mathcal{D},$$

which implies that in expectation, for $\pi_{\widetilde{\pi}^\star}$, we have that for all $i \in \mathcal{D}$:

$$\mathbb{E}^{\pi_{\widetilde{\pi}^\star}}\left[\sum_{h=1}^{H} c_i(s_h, a_h)\right] = \sum_{h=1}^{H}\sum_{(s,a)} \rho_{\pi_{\widetilde{\pi}^\star}}(s, a, h)c_i(s, a) = \sum_{k=1}^{K}\sum_{h=1}^{H}\sum_{(s,a)} \rho_{\widetilde{\pi}^\star}(s, a; h)c_i(s, a)/K \le B_i/K.$$

This means that $\pi_{\widetilde{\pi}^\star}$ is a feasible solution of the hard-constraint program.

Similarly, we have that the expected per-episode total reward of $\widetilde{\pi}^\star$ is the same as the expected total reward of $\pi_{\widetilde{\pi}^\star}$:

$$\mathbb{E}^{\pi_{\widetilde{\pi}^\star}}\left[\sum_{h=1}^{H} r_h(s_h, a_h)\right] = \frac{1}{K}\mathbb{E}^{\widetilde{\pi}^\star}\left[\sum_{k=1}^{K}\sum_{h=1}^{H} r_{k,h}\right].$$

Hence, due to the optimality of $\pi^\star$, we immediately have:

$$\mathbb{E}^{\pi^\star}\left[\sum_{h=1}^{H} r_h\right] \ge \mathbb{E}^{\pi_{\widetilde{\pi}^\star}}\left[\sum_{h=1}^{H} r_h\right] = \frac{1}{K}\mathbb{E}^{\widetilde{\pi}^\star}\left[\sum_{k=1}^{K}\sum_{h=1}^{H} r_{k,h}\right].$$

□

Since our approach incurs sublinear regret with respect to $\pi^\star$, it follows from the above lemma that it incurs sublinear regret with respect to OPT – the total reward across $K$ episodes from the best dynamic policy.

## E  Experimental details

In the experiments, both APPROPO and RCPO use the same policy gradient algorithm, specifically, Advantage Actor-Critic (A2C) Mnih et al. (2016) as the learning algorithm. We implemented CONRL using two version of LAGRCONPLANNER (see algorithm 2 below) as CONPLANNER in which the planner is either value iteration (exact planner) or A2C (approximate planner similar to Dyna model-base RL Sutton (1991)) using fictitious samples. All three algorithms have outer-loop learning rates which we tuned while hyperparameters used for A2C is same across all three methods. Here, we report the result for the best learning rate for each method.

### E.1 LAGRCONPLANNER

Our theoretical results posit that CONPLANNER is solved optimally, which can be indeed achieved via linear programming (see Appendix A). However in our experiments it suffices to use a general heuristic for CONPLANNER. Our approach is to Lagrangify the constraints, and create a min-max mathematical program with the Lagrangean objective:

$$\min_{\forall i \in \mathcal{D}: \lambda(i) \leq 0} \max_{\pi} \left( \mathbb{E}^{\pi, p^{(k)}} \left[ \sum_{h=1}^{H} r^{(k)}(s_h, a_h) \right] + \sum_{i \in \mathcal{D}} \lambda(i) \left( \mathbb{E}^{\pi, p^{(k)}} \left[ \sum_{h=1}^{H} c^{(k)}(s_h, a_h, i) \right] - \xi(i) \right).$$

Define pseudo-reward $r_\lambda^{(k)}$ as

$$r_\lambda^{(k)}(s, a) = r^{(k)}(s, a) + \sum_{i \in D} \lambda(i)[c^{(k)}(s, a) - \xi(i)]$$

With a fixed choice of Lagrange multipliers $\{\lambda(i)\}_{i \in \mathcal{D}}$, this is an unconstrained *planning* program which we refer to as $\text{PLANNER}(p^{(k)}, r_\lambda^{(k)})$ and it can be solved by a planning oracle.

We update Lagrange multipliers via projected gradient descent Zinkevich (2003). The overhead of CONPLANNER is computational, as we do not require new samples. The full procedure is in Algorithm 2. The near-optimality of Algorithm 2 can be proved by leveraging the fact that we are iteratively updating $\pi$ and $\lambda$ using no-regret online learning procedure (Best Response for $\pi$ and OGD for $\lambda$) (e.g., Cesa-Bianchi and Lugosi (2006)). We omit the analysis for Algorithm 2 as it is not the main focus of this work.

---

**Algorithm 2** Lagrangean-based Constrained Planner (LAGRCONPLANNER)

---

1: **hyper-parameters:** learning rate $\eta$
2: **Input:** Estimates $\widehat{p}_k, \widehat{r}_k, \widehat{c}_k$ and bonus $\hat{b}_k$
3: **Compute bonus-enhanced model** $\mathcal{M}^{(k)} = (p^{(k)}, r^{(k)}, \boldsymbol{c}^{(k)})$

$$p^{(k)}(s'|s, a) = \widehat{p}_k(s'|s, a) \quad \forall s, a, s'$$

$$r^{(k)}(s, a) = \widehat{r}_k(s, a) + \widehat{b}_k(s, a) \quad \forall s, a$$

$$c^{(k)}(s, a, i) = \widehat{c}_k(s, a, i) - \widehat{b}_k(s, a) \quad \forall s, a, i \in \mathcal{D}$$

4: Initialize Lagrange parameters $\lambda_1(i) \leftarrow 0$ for $i \in \mathcal{D}$
5: **for** Iteration $k$ from 1 to $N$ **do**
6:     Define
$$r_\lambda^{(k)}(s, a) = r^{(k)}(s, a) + \sum_{i \in D} \lambda(i)[c^{(k)}(s, a) - \xi(i)]$$
7:     $\pi_k \leftarrow \text{PLANNER}(p^{(k)}, r_\lambda^{(k)})$
8:     $\lambda_{k+1}(i) \leftarrow \min \left\{ 0, \lambda_k(i) - \eta \mathbb{E}^{\pi_k, p^{(k)}} \left[ \sum_{h=1}^{H} [c^{(k)}(s_h, a_h, i)] - \xi(i) \right] \right\} \quad \forall i \in \mathcal{D}$
9: **end for**
10: **Return** mixture policy $\pi := \frac{1}{N} \sum_{k=1}^{N} \pi_k$

---

In our experiments, two versions of PLANNER have been implemented: Value Iteration (exact planner) and A2C with fictitious samples (approximate planner)

**Value Iteration as PLANNER**    This program takes $p$ and $r$ as input. Finite horizon value iteration is simply solving the following acyclic dynamic program.

$$Q(s, a, h) = \begin{cases} 0 & h = H + 1 \\ r(s, a) + \sum_{s'} \left[ p(s'|s, a) \max_{a'} Q(s', a', h + 1) \right] & h = 1, \ldots, H \end{cases}$$

then the optimal policy for step $h$ is computed as

$$\pi_h(s) = \text{argmax}_a Q(s, a, h)$$

and the algorithm returns the $H$-step policy

$$\pi = (\pi)_{h=1}^{H}$$

**A2C with fictitious samples as PLANNER** This program takes $p$ and $r$ as input, then, using model $p$ and $r$ it generates episodes and use those samples to train our A2C agent. Since we only call this subroutine with our estimated model ($p \leftarrow \hat{p}$ and $r \leftarrow \hat{r}$) those episodes are fictitious (not adding to sample complexity). The algorithm is given Algorithm 3 (Parameterized policy $\pi_\theta$ and value function estimate $V_\theta$)

---

**Algorithm 3** A2C planner with fictitious samples

---

1: **hyper-parameters:** learning rate $\eta$, $\alpha \in [0, 1]$
2: **Input:** transitions $p$, reward function $r$
3: Define A2C loss

$$L(\theta) = \mathbb{E}^{\pi_\theta, p}[\sum_{h=1}^{H} - \log \pi_\theta(a_h|s_h)(R(h) - V_\theta(s_h)) + \alpha(R(h) - V_\theta(s_h))^2]$$

$$R(h) = \sum_{h'=h}^{H} r(s_h, a_h)$$

4: Initialize $\theta$ arbitrarily
5: **for** Iteration $i$ from 1 to $T$ **do**
6:    Emulate an episode by running $\pi_\theta$ on MDP with transitions $p$ and reward function $r$
7:    update $\theta \leftarrow \theta - \eta \nabla_\theta L(\theta)$
8: **end for**
9: **Return** $\pi_\theta$

---

### E.2 Hyperparameter Tuning

Both CONRL-A2C and RCPO used the Adam optimizer. For our method we performed a hyper-paramter search on both domains over the following values in Table 1; selected values are given in Table 2. Note that reset row refers to when using the A2C planner during each call to the planner we tried the following options: (warm-start) reuse previous weights and reset the optimizer (warm -start), or (continue) continue learning using the previous weights (continue) and optimizer, or (none) reset the model weights and optimizer.

Table 1: Considered Hyperparameters

| Hyperparameter | Values Considered |
|---|---|
| A2C learning rate | $10^{-2}, 10^{-3}, 10^{-4}$ |
| lambda learning rate | $10^0, \{1, 2, 5\} \times 10^{-1}, 2 \times 10^{-2}, 10^{-3}, 2 \times 10^{-3}$ |
| reset | warm-start, continue, none |
| conplanner iterations | $10, 20, 30, 50, 100, 150, 200, 250$ |
| A2C Entropy coeff | $10^{-3}$ |
| A2C Value loss coeff | $0.5$ |

Table 2: Selected Hyperparameters

| Hyperparameter | Gridworld | Box |
|---|---|---|
| A2C learning rate | $10^{-3}$ | $10^{-3}$ |
| lambda learning rate | $2 \times 10^{-1}$ | $10^{-2}$ |
| reset | none | none |
| conplanner iterations | 10 | 10 |
| A2C Entropy coeff | $10^{-3}$ | $10^{-3}$ |
| A2C Value loss coeff | $0.5$ | $0.5$ |

#### E.2.1 TFW-UCRL2

We used the code provided by the author (with no algorithmic parameter changed). Moreover, TFW-UCRL2 uses weights $(L_0, L_1, \ldots, L_k)$ in the objective function $g(w)$ defined in Equation 1 in Cheung (2019). We only tuned these weights to identify the one maximizing the reward while

guaranteeing the constraint satisfaction (for a more fair comparison with the baseline). In our experiments, we have $k = 2$ and you can see the performance of TFW-UCRL2 for $L_0 = 1$ and $L_1 \in \{10^{-2}, 10^{-3}, 10^{-4}, 10^{-5}\}$ in Figure 2.

Figure 2: Performance of TFW-UCRL2 with different choices of $L_1$ ($L_0 = 1$)

# F    Concentration tools

This section contains general concentration inequalities that are not tied with the constrained RL setting considered in the paper.

**Lemma F.1** (Hoeffding). *Let $\{X_i\}_{i=1}^N$ be a set with each $X_i$ i.i.d sampled from some distribution and $\mathbb{E}[X_i] = 0$ for all $i$ and $\max_i |X_i| \le b$. Then with probability at least $1 - \delta$, it holds that:*

$$\left| \frac{1}{N} \sum_{i=1}^N X_i \right| \le b\sqrt{\frac{2\ln(2/\delta)}{N}}.$$

**Lemma F.2** (Anytime version of Hoeffding). *Let $\{X_i\}_{i=1}^\infty$ be a set with each $X_i$ i.i.d sampled from some distribution and $\mathbb{E}[X_i] = 0$ for all $i$ and $\max_i |X_i| \le b$. Then with probability at least $1 - \delta$, for any $N \in \mathbb{N}^+$, it holds that:*

$$\left| \frac{1}{N} \sum_{i=1}^N X_i \right| \le b\sqrt{\frac{2\ln(4N^2/\delta)}{N}}.$$

*Proof.* We first fix $N \in \mathbb{N}^+$ and apply standard Hoeffding (Lemma F.1) with a failure probability $\delta/N^2$. Then we apply a union bound over $\mathbb{N}^+$ and use the fact that $\sum_{N>0} \frac{\delta}{2N^2} \leq \delta$ to conclude the lemma. $\qquad\square$

The following lemma is used when bounding the final regret in the above analysis where we bound the difference between the cumulative Bellman error along the empirical trajectories and the cumulative Bellman error under the expectation of trajectories (the expectation is taken with respect to the policies generating these trajectories cross episodes).

**Lemma F.3.** *Consider a sequence of episodes $k = 1$ to $K$, a sequence of policies $\{\pi_k\}_{k=1}^K$, and a sequence of functions $\{f_k\}_{k=1}^K$ with corresponding filtration $\{\mathcal{F}_k\}$ with $\pi_k \in \mathcal{F}_{k-1}$ and $f_k \in \mathcal{F}_{k-1}$. Each policy $\pi_k$ generates a sequence of trajectory $\{s_{k;h}, s_{k;h}\}_{h=1}^H$. Denote a function $f_k : \mathcal{S} \times \mathcal{A} \to [0, 3H]$, with $f_k \in \mathcal{F}_{k-1}$. With probability at least $1 - \delta$, for any $K$, we have:*

$$\left| \sum_{i=1}^K \sum_{h=1}^H f_k(s_{k;h}, a_{k;h}) - \sum_{k=1}^K \mathbb{E}^{\pi_k} \left( \sum_{h=1}^H f_k(s(h), a(h)) \right) \right| \leq H^{1.5} \sqrt{2 \ln(4K^2/\delta)K}.$$

*Proof.* Denote the random variable $v_{k;h} = f_k(s_{k;h}, a_{k;h})$. Denote $\mathbb{E}_{k;h}$ as the conditional expectation that is conditioned on all history from the beginning to time step $h$ (not including step $h$) at episode $k$. Note that we have: $\mathbb{E}_{k;h}[v_k] = \mathbb{E}^{\pi_k}(f_k(s_{k;h}, a_{k;h}))$. Note that $|v_{k;h}| \leq 3H$ for any $k, h$ by the assumption on $f_k$. Hence, $\{v_{k;h}\}_{k,h}$ forms a sequence of Martingales. Applying Hoeffding's inequality, we have with probability at least $1 - \delta$,

$$\left| \sum_{k=1}^K \sum_{h=1}^H v_{k;h} - \sum_{k=1}^K \mathbb{E}^{\pi_k} \left( \sum_{h=1}^H f_k(s(h), a(h)) \right) \right| \leq 3H \sqrt{2 \ln(2/\delta)KH} = 3H^{1.5} \sqrt{2 \ln(2/\delta)K}.$$

Assigning failure probability $\delta/k^2$ for each episode $k$ and using a union bound over all episodes conclude the proof. $\qquad\square$

## Footnotes

[11]The reason why we sum until $2H$ in the first term is since we want to consider all such visitations that occur in an episode that started with $N_k(s, a) < H$; the additional factor of 2 in the second term comes since, $j/N_t(s, a) \leq 2$ if $N_t(s, a) \geq H$ and the $j$-th visitation happens within the same episode.