[Reviews · NeurIPS 2020]

Review 1

Summary and Contributions: This paper proposed a theoretical RL framework to solve constrained MDP problems under tabular and episodic settings. The distinguishing characteristic is that the paper focused on sample-efficient exploration, which is rarely related with constrained MDPs before.

Strengths: Overall this paper has a great motivation, and provides a thorough mathematics reasoning for the proposed model. This could be the start of a novel research area for solving constrained MDPs.

Weaknesses: My major concerns: 1. line 248 suggested linear programming could be used in ConPlanner, but instead the experiment tested on different unconstrained RL planners under Lagrangian heuristic. I think the papers should have compared results of different constrained problem solver. 2. The paper gave a formulation for knapsack setting, but didn’t test it on any real-world problems. While theoretical proof was plenty, the paper didn’t provide any empirical support, making this method less intuitive. 3. Although the paper claimed they compared the proposed framework with other concave-convex approaches, the problems they experimented on didn’t seem to be concave-convex. Grid world problem such as Mars rover applied in the paper has linear constraints instead of convex ones. On the other hand, in line 270, the paper suggested most previous methods are limited to linear constraints except one. Understandably, It is hard to find previous approaches or benchmarks to compare with under concave-convex settings. The paper should just state them as they were. 4. The vanilla reward function is simply the expected sum of reward, aka, f is nothing but an identity function. Is there any scenario we need to use a complex f function? The paper doesn't seem to offer such examples in the experiment part. 5. In the related work section, the paper mentions that the closest work are (Singh et al., 2020; Efroni et al., 2020; Qiu et al., 2020; Ding et al., 2020). As pointed out, one key difference is that these concurrent work are restricted to linear obj and linear constraint case. A more detailed comparison is welcomed to clarify. For example, what will the result be like when this paper reduces to linear obj and linear constraint? Is there any possibility of extending the concurrent papers' analysis methodology to the convex-concave case? Some minor problems: line 126&178: it would be better if paper add simple statement at where modularity comes in play for both basic and concave-convex setting line 202: in Theorem 4.1 it would be better if there is an explanation of what Ld is. line 206: it would be clearer if paper briefly explains how sublinear aggregate reward regret is related to policy being optimal. line 257: there should not be any reward when rover crashes into a rock. I believe it is a typo. line 269: a typo”previous”

Correctness: The claims, method, and empirical studies look reasonable to me.

Clarity: The paper is well-written.

Relation to Prior Work: As discussed above, the comparison with concurrent related work should be elaborated.

Reproducibility: Yes

Additional Feedback:


Review 2

Summary and Contributions: Authors considered a constraint MDP problem in episodic setting. They proposed a model based method, that learns the transition, reward, constraint with MLE estimate, and form an optimization problem with it. For strategic exploration they used OFU and given the concentration inequalities they proved optimism. Further they extended their algorithm to convex-concave and knapsack setting. It's important to note that the method is for tabular setting and optimization problem is convex in occupancy measures, so efficiently solvable. Additionally they performed some experiments.

Strengths: Theory : paper is very strong theoretically (although I have some question later), proofs are interesting and for convex-concave and knapsack setting non-trivial. However I found the proofs of basic algorithm fairly trivial. Motivation : I strongly believe this is a very important problem, and of value to the community, and has been an under-explored part of the field.

Weaknesses: Writing : I find the paper hard to follow. In order to understand the paper I had to go back and forth to the appendix. I believe authors can put the experiments in the appendix (where for me is of lower importance) and spend more time describing the proofs in the main text and make the intuition clear. (But I understand that field at this point requires a lot of experiments, so I wait to hear other reviewers perspective before making a suggestion here but for me this paper is more of a theoretical contribution)

Correctness: I checked the proofs, and I believe they are correct to the best of my knowledge.

Clarity: As I mentioned, I think writing can be improved, and having to go back and forth to appendix is not optimal. More intuition on how proofs are done is needed in the main text. For example "We apply mean-value theorem to show that this is the case" : This gives almost no intuition on how the proof has been written.

Relation to Prior Work: I am not an expert in this field, but to the best of my knowledge authors have done a good job connecting the dots from bandits literature.

Reproducibility: Yes

Additional Feedback: I have some additional questions regarding theory, 1. PAC bound : I wonder why authors don't represent a PAC bound? Is there a theoretical road block, and if so what needs to be proven for that, and why is it hard? 2. Knapsack Solver: I couldn't find the solver for this, is it also going to be convex? if not, that's a main lacking point of the paper. 3. Scalability : As I said, I believe theoretical analysis by themselves are worth being put out for the community, but I wonder if authors thought about non-tabular setting? And how to scale up the method to larger state spaces, as most practical aspect of the work are not tabular. ## Read authors feedback, and I keep my score as it was.


Review 3

Summary and Contributions: The paper studies an online MDP problem with concave rewards and convex constraints in an episodic setting, where the model parameters on the rewards and resource consumption are not known. The authors proposed an optimism based algorithm, and quantify the performance guarantee in terms of the reward regret and the constraint regret. The main result is sqrt{K} expected regret bounds for those two regrets. The authors also studied a hard constraint version of the problem, where the resource constraints must be satisfied always.

Strengths: The results incorporate general concave rewards and convex constraints, which provide a general model to capture a variety of applications. Personally, I feel that the notion of aggregate rewards for RL has been long neglected, in view of its numerous applications in RL exploration. This work fills in the gap in the case of episodic MDP, which makes myself to be slightly in favor of acceptance than reject. The regret bounds essentially match those for the scalar reward setting, and the authors also point out the non-trivial steps in generalizing from (Agrawal and Devanur 2011) to the current episodic MDP setting.

Weaknesses: 1. While in the "Strengths" I remarked that regret bounds essentially match those for the scalar reward setting, I feel that there is still some gap in the bound. In Theorem 4.1, the consumption regret upper bound of "Ld · CONSREG" has a rather loose dependence on d, as in it does not match the dependence of \| 1_d \| in (Agrawal and Devanur 2011). Here, I denote 1_d as the d-dimensional all 1 vector, and g is L Lipschitz continuous with respect to the norm \|\cdot\|. 2. In the main results, the authors show that the constraint regrets are bounded in expectation. Do these bound translate to a high probability bound, in the same way as (Badanidiyuru 2013, Agrawal and Devanur 2014, Cheung 2019)? 3. Another place that needs substantial improvement is the numerical experiments. While the authors have provided additional details for the numerical experiments in Appendix E, there are quite a few places that require clarification: - How does the plots in Figure 1 corroborate with the regret bounds? More precisely, it is not clear how a trajectory means in the online model. For example, if I look at the top left plot for RCPO, it reports that at No. of trajectories = 500, the reward is \approx 1.65. What does it mean? Does it mean that if I run the RCPO with 500 episodes than the empirical average reward realizes as 1.65, or does it mean something else? - Can the authors provide plots about the cumulative regret of the algorithms (at least for the proposed algorithms, in case they don't make sense for existing algorithms like RCPO for some reason)? This provide a more direct way to empirically evaluate the proposed algorithms. - In relation to the previous point, in footnote 7, it remarks that the bottom row corresponds to "the aggregate actual constraint incurred during training". However, in an online model, how is the notion of "during training" defined? - It is not clear why the authors include A2C, which requires the access to the latent model on p as shown in the Appendix (so it seems to violate the model assumption of not knowing p for example?) - I am in fact quite confused by column C. Is there any reason why TFW-UCRL2 is run with significantly more trajectories than the other two algorithms ConRL-A2C and ConRL-Value Iteration? - When I tried to dig deeper in Appendix E, it is stated that “TFW-UCRL2 gives fixed weights to reward and constraint violation and maximizes a scalar function. Therefore we tuned TFW-UCRL2 for different weights and the reported result is for the best weight.” Nevertheless, to my knowledge, the TFW-UCRL2 in fact assign dynamic weights () to the penalty function for the constraints, and those weights do not need tuning in the sense that the dynamic update of the weights Can the authors provide a high level sketch on how they implemented the TFW-UCRL2 in the online episodic setting?

Correctness: The theoretical results are correct to my knowledge. I am not entirely sure about empirical methodology, and I look forward to the reply by the authors on the questions in "Weakness".

Clarity: Apart from the numerical results, other parts are well written. I strongly advice the authors to provide more details (at least enough details to address the questions in the "Weakness") in the appendix? Some minor comments for polishing the paper: Line 192: The convexity in occupation measure might not be immediately clear to readers unfamiliar with (Agrawal and Devanur 2011). The authors could make reference to the relevant Lemma in that reference. Line 269: prevnous -> previous

Relation to Prior Work: To my knowledge, the immediately relevant literature, in particular the concurrent results (Singh et al., 2020; Efroni et al., 2020; Qiu et al., 2020; Ding et al., 2020), are clearly discussed. The authors could also include the following reference that has a similar setting to (Cheung 2019): Jean Tarbouriech, Alessandro Lazaric. Active Exploration in Markov Decision Processes. AISTATS 2019.

Reproducibility: Yes

Additional Feedback: In the concave-convex setting in Section 4: The Lagrangian heuristics LAGRCONPLANNER considered in the numerical experiments appear much more tractable than the basic ConRL algorithm, I feel that the authors could consider bounding the regret of those heuristics directly. In fact, LAGRCONPLANNER bears a lot of similarity to the dual based approach by Agrawal and Devanur 2011. ########### Post Rebuttal ############## The authors have addressed the major concerns on the numerical experiments, and the overall score is increased.I look forward the authors to provide those details in the appendix for the numerical experiments.


Review 4

Summary and Contributions: The paper proposes an algorithm to learn policies in environments with concave rewards and convex constraints, in the tabular and episodic setting. The authors provide theoretical guarantees on the performance w.r.t reward and consumption regrets. They also demonstrate the algorithm on 2 environments - box and mars rover with comparison against constrained policy optimization baselines. I've read the rebuttal and going to stick to my rating.

Strengths: The problem statement is clearly defined. The assumptions are clearly laid out. The objectives of bounding reward and consumption regrets seem fair. I haven't verified the theoretical analysis of the claims, so can't comment on that.

Weaknesses: The experimental setup could improve: 1. The environments chosen don't adequately demonstrate the resource constrained setup that they wish to deploy the algorithm in. Specifically, the knapsack setting is interesting but a challenging environment is lacking. A real world example is that of budgets earmarked for campaigns or energy resources in games. 2. The tabular setting is helpful for theoretical analysis but it would be helpful to mention how the algorithm and analysis would translate to the function approximation case.

Correctness: Not experienced in this area and hence couldn't verify the claims.

Clarity: The paper is generally well written. Nit: Line 104: Q-function is known as the action-value function in the standard literature.

Relation to Prior Work: The related work on bandits is mentioned. The algorithmic improvements appear less novel while the analysis is thorough.

Reproducibility: Yes

Additional Feedback:

[Author Response · NeurIPS 2020]

We thank all reviewers for their thoughtful feedback that can help enhance the presentation of our results. Below we
respond to the major points raised by the reviewers (for each point, we refer to the particular reviewers that raised it).

**ConPlanner via unconstrained RL planners under Lagrangian heuristic in experiments (Reviewer** 1**).** The
baselines (RCPO and ApproPO) are based on unconstrained RL solvers under Lagrangian heuristic. To enable a more
fair comparison to the baselines, we implemented a ConPlanner that is as close to this as possible (implementing it with
an unconstrained planner under Lagrangian heuristic). This provides an approximately optimal planner (rather than
an optimal constrained planner such as LP) but our theorems are anyway approximation-preserving (this is direct in
our proofs but we can add a note about that). As a result, by having an implementation that is close to the one of the
baselines, we could better demonstrate the statistical improvement of our methodology.

**Lack of concave-convex and knapsack experiments (Reviewer** 1**).** Benchmarks do not currently exist for con-
vex/knapsack constraints. For the sake of transparency, we decided to compare against prior works handling concave-
convex or knapsack settings on established benchmarks (for the linear case) rather than come up with our own (although
it renders the empirical comparison a bit more limited). We will clarify this decision (as the reviewer recommends).

**Comparison to concurrent work (Reviewer** 1**).** The results of the concurrent works are similar to our warm-up basic
setting (Section 3). Although we believe that their techniques can extend to more general settings, we do not see a way
to do that without using the new technical tools we create in Sections 4 and 5 (e.g., see next point re mean-value).

**Intuition behind mean-value theorem (Reviewer** 2**).** The mean-value theorem is applied in two places: a) proving
that the optimal policy is feasible for our constrained planner (lines 520-525 of the appendix) and b) to enable the regret
decomposition (lines 535-538 of the appendix). The latter application draws a distinction to the multi-armed bandit
setting as it is required to address the difference between the model used and the true model — Eq. (20) holds trivially
when $H = 1$. As suggested by the reviewer, we will add a discussion about that in the main body as it provides intuition
of where mean-value applies and also draws an elegant technical distinction to the bandit setting.

**PAC bound (Reviewer** 2**).** A regret bound can be transferred to a PAC bound: a $\sqrt{K}$ regret bound can be turned into a
$1/\epsilon^2$ PAC bound by taking the resulting mixture policy. We will add a note in the final version.

**Knapsack solver (Reviewer** 2**).** The knapsack solver is provided in Appendix A.3 and is a linear program with
occupation measures as variables (thus it is optimally solvable).

**Beyond tabular settings (Reviewers** 2 **and** 4**).** Our results can serve as starting point towards extending to non-tabular
settings such as Linear or Lipschitz-continuous MDPs where optimistic algorithms for the unconstrained counterparts
exist. We will discuss the additional challenges that arise in these settings and explicitly state them as future directions.

**Dependence on d and high probability (Reviewer** 3**).** We assumed function $g$ (line 181) is $L$-Lipschitz continuous
under $L_1$-norm ($\|1_d\|_1 = d$ here). More generally, our analysis extends when $g$ is $L$-Lipschitz with respect to $L_p$ norm
($p \geq 0$) and obtains $L \cdot \|1_d\|_p$ dependence. Our results also extend to high-probability by an application of Lemma F.3.

**Clarifications on the plots (Reviewer** 3**).** The number 1.65 says that, after running the setting for 500 episodes (or
trajectories) each with $H = 30$ steps, the expected reward of the resulting policy $\pi_k$ at episode $k$ is equal to 1.65. Note
that this is not the time-average reward; the plot for cumulative regret looks similar (note that the $x$-axis is in log-scale,
we will add it as the reviewer suggests). The second row corresponds to expected cost of policy $\pi_k$, i.e., $\mathbb{E}^{\pi_k}[\sum_h c_h^\star]$.
The third row corresponds to the cumulative empirical cost, i.e., $\sum_{t=1}^k \sum_{h=1}^H c_{t,h}$ (the term *during training* meant to
show what happens if we terminate at episode $k$ but we will replace this term by the exact formula to make it clear).

**Regarding access to the latent model (Reviewer** 3**)** Algorithm 3 is called with the empirical model transitions $\hat{p}_k$ as
a parameter – it does not require access to the latent model; it only uses the model estimates (we will clarify this).

**Comparison to TFW-UCRL2 (Reviewer** 3**).** In the experiments, our goal was to find a (constraint-feasible) policy
that guarantees a desired reward in as few episodes (trajectories) as possible. In the plots, we display the number of
trajectories needed in order for this goal to be achieved (TFW-UCRL2 requires more trajectories to achieve that). We
will add a column about the alternative metric of regret – we ran it and it looks qualitatively similar (that said, it is
probably useful to show pictorially the vanishing regret property so such an addition would be useful for the paper).

Regarding how we ran TFW-UCRL2, we use the code provided by the author of TFW-UCRL2 (with no algorithmic
parameter changed). TFW-UCRL2 uses weights $(L_0, \ldots, L_K)$ in the objective function $g(w)$ (Eq. 1 page 2 of that paper
in its NeurIPS version). We only tuned these weights to identify the ones maximizing the reward while guaranteeing
constraint satisfaction (for a more fair comparison to the baseline). We will include the additional plots for different
values of $(L_0, \ldots, L_K)$ in appendix.

[Meta-Review · NeurIPS 2020]

While it is true that constraints can typically be made part of the normal optimisation process in RL, by encapsulating them into the reward function, it can often be much easier to specify constraints directly, which is the setting this paper considers. The reviewers were positive about the motivation and execution of this paper, and were all in favour of accepting the paper. I would suggest already motivating this setting, at least somewhat, in the abstract, to help interesting readers find and appreciate this paper more easily.